# Hierarchical Reinforcement Learning for Sparse-Reward Search in Commutative Algebra

Giorgi Butbaia [1]   Paul Orland [1]   Coco Huang [2]   Davide Passaro [1]   Lucas Fagan [1]   Michele Tarquini [1]
Hailong Dao [3]   David Eisenbud [4]   Ali Shehper [1]   Sergei Gukov [1]

## Abstract

Applying machine learning techniques to solving long-standing mathematical conjectures can be particularly challenging due to their extreme reward sparsity. As an illustrative example, we consider Kalai's algebraic Hirsch conjecture and recast the construction of its counterexamples as a sparse-reward reinforcement learning problem on graphs. We propose a constrained options-based HRL framework with an equivariant graph neural network policy, which allows us to learn useful temporal abstractions for this task. We evaluate our approach over a wide range of degrees and demonstrate that it consistently outperforms classical RL algorithms as well as greedy search. By exploiting the hierarchical structure of the problem, we effectively provide a first-of-its-kind application of HRL to a problem in commutative algebra.

## 1. Introduction

Reinforcement learning has emerged as a powerful tool for tackling combinatorial search problems in mathematics, from discovering matrix multiplication algorithms (Fawzi et al., 2022) to generating counterexamples in geometry (Swirszcz et al., 2025). However, many fundamental problems in pure mathematics present an extreme challenge: the objects being sought are so rare that random exploration almost never encounters them, leading to reward signals that are essentially zero throughout training. This paper addresses one such problem in commutative algebra — the search for rare monomial ideals that violate an algebraic

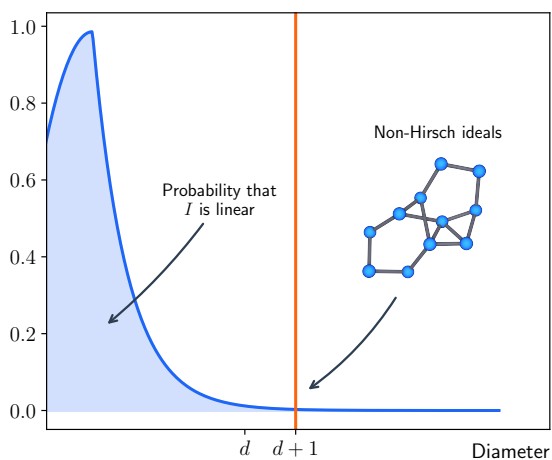

*Figure 1.* Probability that an ideal $I$ is linear as a function of the diameter $\mathrm{diam}(I)$. The probability rapidly decays as the diameter approaches $d + 1$, which is where non-Hirsch ideals live.

analogue of the Hirsch diameter bound — and demonstrates that hierarchical reinforcement learning (HRL), when combined with domain-specific constraints, can succeed where standard RL methods fail entirely.

The Hirsch conjecture, proposed in 1957, concerns the diameter of polytope graphs and has deep connections to the complexity of the Simplex algorithm for linear programming (Dantzig, 1963). While the original conjecture was disproven after 50 years (Santos, 2012), an algebraic generalization due to Kalai remains open. This algebraic version asks whether the diameter of graphs associated with *linearly-presented monomial ideals*[1] can exceed the degree of their generators — objects we call *non-Hirsch ideals* — and by how much. Finding such ideals is extremely difficult: as illustrated in Figure 1, the probability that a random ideal satisfies both the linearity and large-diameter conditions decays rapidly, making this a canonical example of a sparse-reward environment.

Our choice of problem is motivated in part by its mathematical significance: commutative algebraists have been

[1]Department of Mathematics, California Institute of Technology, Pasadena, CA [2]Department of Mathematics, Temple University, Philadelphia, PA [3]Department of Mathematics, University of Kansas, Lawrence, KS [4]Department of Mathematics, University of California, Berkeley, Berkeley, CA. Correspondence to: Giorgi Butbaia <gbutbaia@caltech.edu>.

*Proceedings of the 43rd International Conference on Machine Learning*, Seoul, South Korea. PMLR 306, 2026. Copyright 2026 by the author(s).

---

[1]In this manuscript we will use linearly-presented monomial ideals and linear monomial ideals interchangeably.

very interested in linearity properties of monomial ideals (linear presentation/resolution/regularity) for a long time, and there is a large literature; see e.g. (Fröberg, 1990; Eagon & Reiner, 1998; Jahan & Zheng, 2010; Dao & Eisenbud, 2022) for a tiny sample.

From the machine learning perspective, our problem shares key characteristics with notoriously difficult RL benchmarks like *Montezuma's Revenge* (Ecoffet et al., 2021): rewards are extremely sparse, the state space grows combinatorially, and success requires discovering a precise sequence of actions. Standard algorithms such as PPO and SAC fail to find any solutions beyond the smallest problem instances (Table 1). This motivates our central question: *can we design an HRL framework that exploits the mathematical structure of the problem to enable effective learning?*

Our key insight comes from analyzing the trajectories of trained agents: successful paths consistently pass through a particular class of intermediate states that we call *spines* — simple graphs satisfying the diameter constraint. This observation suggests a natural temporal abstraction: first construct a spine, then "linearize" it by adding generators to satisfy the algebraic condition. We formalize this as a *Chained Constrained Options* framework, where two options are executed in sequence, each with intra-option policy constraints that prune invalid actions while preserving the ability to reach all solutions.

This framework addresses several well-known challenges in HRL. The instability that typically arises when learning multiple policy levels simultaneously (Levy et al., 2019) is mitigated by our constraint structure, which provides stable sub-goal definitions. Unlike methods that struggle to learn meaningful sub-goals end-to-end (Vezhnevets et al., 2017; Nachum et al., 2018), our mathematically-motivated decomposition yields interpretable intermediate states. The constraints themselves act as a form of curriculum, guiding exploration toward the sparse region of state space where solutions exist.

**Contributions.** We make the following contributions:

- We formulate sparse-reward counterexample search for non-Hirsch ideals as a graph-based RL environment, and provide strong baselines from both RL and classical search.

- We propose a *Chained Constrained Options* HRL framework that leverages empirically identified bottlenecks (spines) and enforces algebraic constraints within options to mitigate HRL instability and subgoal-specification issues.

- We design an equivariant graph neural policy with *syzygy-aware message passing* and obstruction features, and show that the resulting agent consistently outperforms

standard RL and greedy search across a range of degrees, representing the first successful application of HRL to a problem in commutative algebra.

**Roadmap.** Section 2 provides minimal mathematical background (more details can be found in Appendix A). Section 3 defines the search and RL formulations and establishes classical search baselines. In section 4 we introduce *syzygy-aware message passing* and describe our graph neural network architecture. Section 5 analyzes why standard RL approaches fail and identifies bottleneck states. Section 6 presents our constrained options HRL framework and experimental results. We conclude in Section 7 with limitations and directions for extending HRL-style agentic search to other problems.

The repository containing additional materials is available at `https://github.com/Math-AI-Caltech/alghirsch-hrl`.

### Conflict of Interest Disclosure

The authors declare no financial conflicts of interest.

## 2. Preliminaries: Algebraic Hirsch Conjecture

The combinatorial version of the Hirsch conjecture is well known to be tractable with computational tools. In fact, one of the first counterexamples was produced via a computer search, generating a polytope with 36,442 vertices (Santos, 2012). Recent approaches, such as (Swirszcz et al., 2025), have successfully utilized machine learning tools to construct a large number of counterexamples.

The algebraic version of the conjecture can be similarly formulated as an extremal problem. To construct counterexamples, one seeks to construct homogeneous square-free monomial ideals $I$, called *non-Hirsch* ideals, that are *linearly-presented*, and satisfy a large diameter constraint:

$$\operatorname{diam}(I) > d, \qquad (1)$$

where $d$ is the degree of the generators of $I$ and $\operatorname{diam}(I)$ is the diameter of the graph $G_I$ associated to the ideal $I$.

Homogeneous square-free monomial ideals $I$ of degree $d$ in $n$ variables can be conveniently characterized by their generators, sets of words of fixed length $d$, consisting of non-repeating (square-free) letters from an alphabet of size $n$. For example, given an alphabet of $n = 3$ letters $x_1$, $x_2$, $x_3$, a homogeneous square-free monomial ideal $I$ of degree $d = 2$ can be constructed by choosing generators of the form $x_i x_j$, for $i \neq j$. A graph $G_I$ can be associated to an ideal $I$ by assigning a vertex to each generator of the ideal $I$, with edges connecting generators that differ by at most one letter. The *linearity* condition is defined using commutative

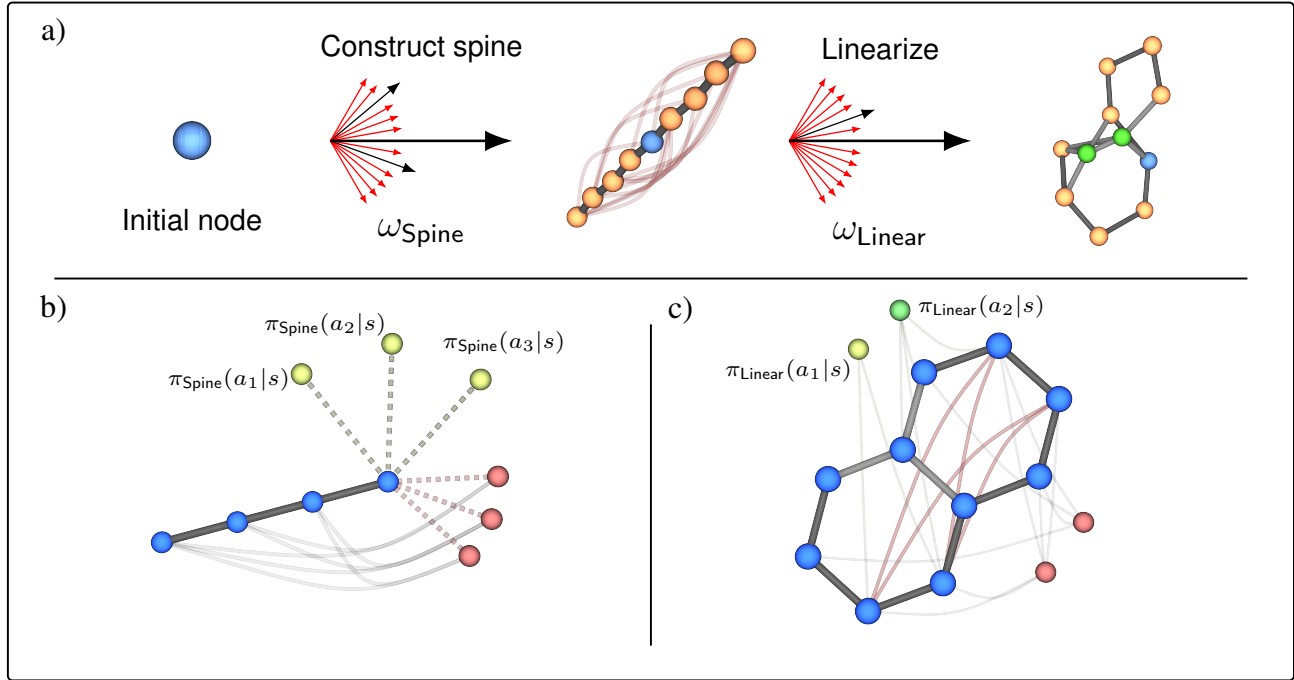

*Figure 2.* The two-step process with constrained options $\omega_{\text{Spine}}$ and $\omega_{\text{Linear}}$ is shown on a). The red nodes in b) and c) denote invalid actions as they violate the intra-option constraints, while yellow and green nodes denote valid and terminal actions, respectively. The intra-option policies for $\omega_{\text{Spine}}$ and $\omega_{\text{Linear}}$ are $\pi_{\text{Spine}}$ and $\pi_{\text{Linear}}$, respectively. The red edges in a) and c) denote irreducible edges. For brevity, we shall use $\pi_S$ (and $\omega_S$) in place of $\pi_{\text{Spine}}$ and $\pi_L$ in place of $\pi_{\text{Linear}}$.

algebra; one can think of linearity as an algebraic complexity measure of the ideal (see Appendix A for details).

Homogeneous square-free monomial ideals satisfying the large-diameter condition and the linearity condition individually (1) are abundant, but satisfying both conditions at the same time is highly non-trivial; ideals that are linear and have a diameter exceeding the degree are exceedingly rare (see Fig. 1). In the RL setting, this scarcity translates to a sparse reward environment, making learning non-Hirsch ideals very challenging.

We conclude this part with a summary of the key mathematical notations used throughout the paper:

| | |
|---|---|
| $I$ | (square-free) monomial ideal |
| $G_I$ | graph associated to $I$ |
| $n$ | number of variables |
| $d$ | degree |
| $\text{diam}(I)$ | diameter (of the graph $G_I$) |

## 3. Search Algorithms for Constructing non-Hirsch Ideals

Our primary goal is to find an optimal search algorithm for linearly-presented ideals satisfying the large-diameter condition (1). To that end, we present two different classes of search strategies:

- **Classical Search**, including greedy-search algorithms such as Best-first search, A*, etc. These strategies are used to establish baselines.

- **Reinforcement Learning**, which learn useful patterns for constructing non-Hirsch ideals. These strategies are the main focus of our work.

We define the state space $\mathcal{S}$ as the set of all monomial ideals of a given degree $d$. We use this state space for both strategy classes. We take the action space $\mathcal{A}$ to be the set of moves which toggle the inclusion of a generator in a state ideal. Since the number of variables is fixed and the monomial ideals are square-free, the state and action spaces have sizes

$$|\mathcal{S}| = 2^{N(n,d)}, \quad |\mathcal{A}| = N(n,d) := \binom{n}{d}, \qquad (2)$$

where $n$ is the number of variables and $d$ is the degree of the monomials.

The combinatorial growth in the degree $d$ of the monomials and in the number $n$ of variables makes brute-force approaches intractable. To amend this growth the following design choices are employed:

**1. Feature representation**: Let $I_{\text{max}}(d, n)$ be the ideal generated by all possible square-free monomials of degree $d$ in $n$ variables. This ideal can be represented by a graph $G_{\text{max}}(d, n)$ with nodes that are decorated with textual fea-

*Table 1.* Performance comparisons of constructing non-Hirsch ideals by different algorithms. The effective success rate is defined as a count of successful environment terminations over the number of environment interactions.

| Algorithm | Type | Effective Success Rate (↑) | | | |
|---|---|---|---|---|---|
| | | $d = 4$ | $d = 5$ | $d = 6$ | $d = 7$ |
| PPO (Schulman et al., 2017) | — | $1.94 \times 10^{-6}$ | ✗ | ✗ | ✗ |
| SAC (Haarnoja et al., 2018) | — | $9.56 \times 10^{-6}$ | ✗ | ✗ | ✗ |
| SAC+HuRL (Cheng et al., 2021) | Spine priors | $11.16 \times 10^{-6}$ | ✗ | ✗ | ✗ |
| SAC+HER (Andrychowicz et al., 2017) | Spine priors | $7.78 \times 10^{-5}$ | ✗ | ✗ | ✗ |
| SAC+Priors | Spine priors | — | $0.15 \times 10^{-6}$ | ✗ | ✗ |
| SAC+Buffer (Swirszcz et al., 2025) | Spine priors | — | $2.27 \times 10^{-6}$ | ✗ | ✗ |
| Best-first search | Brute-force | $2.03 \times 10^{-2}$ | $1.55 \times 10^{-3}$ | $6.59 \times 10^{-4}$ | $3.9 \times 10^{-4}$ |
| **Constrained Options** | **HRL** | $\mathbf{1.08 \times 10^{-1}}$ | $\mathbf{8.74 \times 10^{-2}}$ | $\mathbf{6.78 \times 10^{-2}}$ | $\mathbf{3.16 \times 10^{-2}}$ |

tures corresponding to the word representations of monomials. The graph $G_I$ corresponding to an ideal $I$ is a subgraph of $G_{\max}(d, n)$.

Motivated by this construction, we represent each ideal with a mask which picks out the nodes from $G_{\max}(d, n)$ corresponding to the generators of the ideal $I$. This representation allows to process the ideals using message-passing algorithms, similar to those of graph attention. We demonstrate that this provides sufficient expressivity to capture relevant features for solving environments, provided that the graph associated to an ideal contains sufficient information.

**2. Environment Shaping**: The graph features associated with an ideal are insufficient to reliably determine whether the ideal is linearly-presented and to predict whether a long-horizon action will produce a linearly-presented ideal.

In Appendix A, we provide an extension of these graphs by augmenting them with *irreducible edges*. The irreducible edges quantitatively measure how far a given ideal is from being linearly-presented, and also provide topological information regarding the location of these obstructions. The topological information can then be picked up by the message-passing protocol, providing crucial information to the RL agents. This information allows us to shape the observations and, through a heuristic $h(I)$, the reward function. We take the heuristic to be

$$h(I) = \left| \left\{ e \in \bigcup_{l > 1} E_l(I) \,\middle|\, e \text{ is irreducible} \right\} \right|, \quad (3)$$

where $E_l$ is the set of edges connecting nodes corresponding to the generators in $I$ which differ by $l$ variables.[2] For an efficient algorithm for checking if a given edge is reducible, see Alg. 1.

**3. Constraints**: Equation (2) shows that the domain on which RL agents operate scales combinatorially. Naive ap-

proaches, which we will discuss in Sec. 5, suffer from poor performance as the agents rarely get any positive reward, even with the aid of the heuristic from Equation (3).

In order to identify useful constraints, we first observe the behavior of the agents and identify the *bottleneck states* (Solway et al., 2014) that lie on the high-probability paths toward non-Hirsch ideals, which exhibit a particularly simple structure. We then introduce a set of *chained constraints* to sequentially prune the state-action spaces by introducing a subgoal leading to the *spines* bottleneck states, without preventing the agent from reaching any valid non-Hirsch ideal. This suggests recasting the problem in the framework of HRL, where the resulting temporal abstractions (Fig. 2) define a useful option set for constructing such ideals.

## 4. Features and Syzygy-aware Message Passing

Any ideal $I$ can be represented by a subgraph $G_I \subseteq G_{\max}(d, n)$. However using the subgraph alone as the state representation causes the model to fail at capturing the long-term effects of introducing new nodes. Introducing new nodes is critical for removing irreducible edges, as irreducibility depends on the existence of paths between the edges' endpoints. Therefore the model requires explicit knowledge of the topology of the full graph $G_{\max}(d, n)$, with all possible generators, in order to capture both procedural and predictive knowledge. By identifying each graph $G_I$ with a subgraph of $G_{\max}(d, n)$, we may represent each state by decorating the nodes of $G_{\max}(d, n)$ with an *inclusion* bit, which indicates if the node is part of the subgraph. We further augment the resulting decorated graph with a set of irreducible edges corresponding to the ideal $I$, ensuring that the network has direct access to the obstructions preventing the ideal from being linearly-presented.

Each action corresponds to flipping the inclusion bit of the corresponding generator. We therefore parametrize both

---

[2]See Appendix A for a more detailed definition of $E_l$.

the policy and the $Q$-function with a GNN which has a single-dimensional output for each node. For the policy, these output features are converted to softmax probabilities, while for the $Q$-function, they provide a direct estimate of expected return of flipping each generator in a given state. This approach ensures that both functions remain equivariant under permutation of the nodes of the graph.

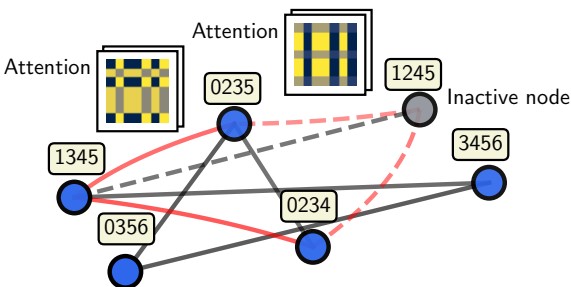

*Figure 3.* Syzygy-aware message-passing architecture used for processing node features. Blue nodes belong to the current state, while the gray nodes do not. The red edges denote the irreducible edges.

Out of two main objectives, computing the diameter requires only the topological information of the graph, whereas verifying if an ideal is linearly-presented further requires the textual information associated to the nodes. From the reduction algorithm 1, it is evident that an effective method for representing the textual information is via binary encoding of generators. In particular, we use the map:

$$x_{i_1} x_{i_2} \ldots x_{i_d} \longmapsto \sum_{j=1}^{d} 2^{i_j - 1}, \qquad (4)$$

and use the binary sequences of length $n$ of the resulting number to represent the features associated to each node. Noting that the reduction algorithm 1, requires pairwise features, which consist of binary logical operations such as & and | on pairs of nodes, we create a message-passing layer, which we call *Syzygy-aware* message-passing (SConv), which leverages cross-attention to compute the aggregation function. In particular, we compute the cross attention for an edge connecting a source node with a target node with binary sequence representations $v^{\mathrm{src}}$ and $v^{\mathrm{tgt}}$ respectively[3]:

$$\mathrm{Aggr}(v^{\mathrm{src}}, v^{\mathrm{tgt}}) = \mathrm{CrossAttn}(v^{\mathrm{src}}, v^{\mathrm{tgt}}, v^{\mathrm{tgt}}). \quad (5)$$

We combine the resulting features with multiple layers of GAT (Veličković et al., 2017) using a GPS transformer (Rampášek et al., 2022). We demonstrate the effectiveness of this model via a brief ablation study in Fig. 6.

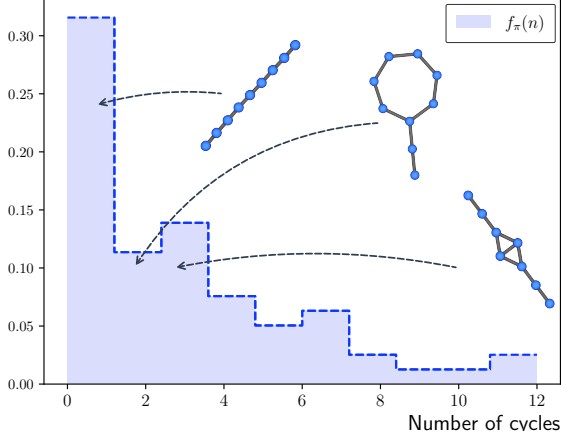

*Figure 4.* Frequency $f_\pi(n)$ in (7) of the simplest graphs encountered in the trajectories sampled via behavioral policy $\pi$ trained using SAC.

## 5. Agentic Search for non-Hirsch Ideals

We formulate the problem of constructing non-Hirsch ideals as an MDP $\mathcal{M} = (\mathcal{S}, \mathcal{A}, p, r, \gamma)$ where the transition probabilities $p$ are non-stochastic $p(s'|s, a) = \delta_{a(s)}(s')$, where $\delta$ is the delta distribution: $\delta_a(b) = 1$ if $a = b$ and 0 otherwise.

Given that an arbitrary state in general is neither linearly presented nor does it have diameter exceeding the degree as in Equation (1), we first consider a "naive" reward function defined as a cumulative penalty of the required conditions: $r(s, a, s') = r_L(s, a, s') + r_D(s, a, s')$ where $r_D(s, a, s') = -1$ if $\mathrm{diam}(s') \leq d$ and 0 otherwise. Similarly:

$$r_L(s, a, s') = \begin{cases} 1 & s' \text{ is linear and } \mathrm{diam}(s') > d \\ 0 & \text{otherwise} . \end{cases} \quad (6)$$

Linearity checks were initially done via Macaulay2 (Grayson & Stillman), which posed a significant computational bottleneck. To circumvent this, we trained a supervised model to predict if a given ideal is linearly presented, which we then used for approximating $r_L$ reward function. We find that both PPO and SAC require large amount of iterations to produce solutions (as indicated in Table 1), rendering them infeasible for scaling. We instead use the results to identify bottleneck states (Solway et al., 2014).

Standard techniques, such as eigenoptions (Machado et al., 2018) and successor representations, use state-visitation statistics to identify the bottleneck states. Instead we focus on identifying the simplest graph in each trajectory since tabular methods are infeasible and we do not have useful feature-level representations for the purposes of eigenoptions. We use the number of cycles $b_1(G)$ as a measure of "complexity" of a graph $G$. Instead of counting the frequency of visitations of a given state (as in SR), we focus

---

[3]See Appendix D.1 for a more detailed definition of SConv.

on the distribution of the least complex graphs over all trajectories. Let $\pi$ be a behavioral policy; we consider the probability that the graph has $n$ cycles along a trajectory sampled via $\pi$, :

$$f_\pi(n) = \mathbb{E}_{\tau \sim \pi} \left[ \mathbf{1} \left( \min_{t \geq 0} b_1(s_t) = n \right) \right] . \qquad (7)$$

Using the trajectories collected with SAC, we construct a Monte-Carlo approximation of the distribution of $f_\pi(n)$ in Equation (7), shown in Fig. 4. Trees and tadpole-like graphs emerge as the most common types of least-complex graphs encountered along the trajectories. This motivates introduction of temporal abstractions which focus solely on constructing these bottleneck states, which we will discuss in-depth in the next section.

## 6. Spines and Temporal Abstractions

While standard RL algorithms with a naive reward function (6) are able to produce some non-Hirsch ideals in lowest non-trivial degree $d = 4$, they fail at producing results for larger degrees. Based on the results, we have identified a prevalence of certain path graphs which appear in the majority of trajectories sampled via the optimized behavior policy. In this section, we first quantitatively assess the effectiveness of the approach and then reformulate the problem within options framework to significantly accelerate the construction of non-Hirsch ideals.

### 6.1. Spines

One of the main conditions for identifying non-Hirsch ideals is the large diameter condition in (1). The results of Section 5 motivate the following definition:

**Definition 6.1.** An ideal $I$ is a *spine* if the graph $G_I$ associated to the ideal is a path graph whose diameter $\mathrm{diam}(I)$ satisfies the large diameter condition in (1).

As part of the ablation study of spines, we evaluate the impact on performance of replacing the prior distribution $p(s_0)$ with a "uniform" distribution of spines $p_S(s_0)$. Uniformly distributed spines are constructed procedurally using random-DFS on the graph $G_{\mathsf{max}}(d, n)$ (as defined in Section 3) with a random-uniform node selection. Instead of the naive rewards in (6), we construct and shape rewards using different approaches.

**1. Step-cost reward**: Given a randomly initialized spine, our goal is to find nearest non-Hirsch ideals. In order to ensure that the optimal value function $V^*$ is equivalent to the cost-to-go function of the environment, we assign a $-1$ penalty for each step, effectively turning the reward function to $r(s, a, s') = -1$. Using the spines $s_0 \sim p_S(s_0)$ as initial states in lieu of $p(s_0) = \mathcal{U}\{0, 1\}^{N(n, d)}$, where $\mathcal{U}\{0, 1\}$ is

the uniform distribution on $\{0, 1\}$, the agents are able to find non-Hirsch ideals of degree $d = 5$.

**2. HuRL ([Cheng et al., 2021](#))**: Since a heuristic (3) can be efficiently computed, we may use it for shaping the reward function $r(s, a) = -1$ in real time. There are two major approaches for potential-based shaping: a PBRS shaping ([Ng et al., 1999](#)), which provides an optimal policy-preserving transformation of the reward function:

$$r(s, a, s') \to r(s, a, s') + \gamma \phi(s') - \phi(s) , \qquad (8)$$

using a potential function $\phi$, and a Heuristic-Guided RL ([Cheng et al., 2021](#)), which transforms the original MDP $\mathcal{M}$ to a $[0, 1] \ni \lambda$-scheduled family of MDPs $\tilde{\mathcal{M}}_\lambda = (\mathcal{S}, \mathcal{A}, p, \tilde{r}_\lambda, \lambda\gamma)$ where:

$$r(s, a, s') \to r(s, a, s') + (1 - \lambda)\gamma \phi(s') , \qquad (9)$$

where $\lambda$ moves from $0$ to $1$. While we've observed some marginal improvement in the performance for $d = 4$, we were not able to substantially improve the performance for higher degrees $d > 4$, as indicated in the Table 1.

**3. Population Buffer ([Swirszcz et al., 2025](#))**: Another technique that leverages the heuristic is to maintain a priority buffer, with priority function given by the heuristic in (3). In ([Swirszcz et al., 2025](#)), the authors describe the "Hopper" algorithm, wherein the procedure involves sampling an element from the priority buffer, attempting to improve it (as measured by improvement in the heuristic) and, if successful, storing the improved state back to the buffer. If the heuristic provides useful intermediate goals, the average score of the buffer increases. For non-Hirsch ideals, we define priority function $P(I)$ as a combination of the number $h(I)$ of irreducible ideals defined in (3) and the diameter $\mathrm{diam}(I)$ of the graph associated to the ideal:

$$P(I) := -h(I) - |\mathrm{diam}(I) - (d + 1)| . \qquad (10)$$

While we observe that this approach outperforms previous techniques by making $d = 5$ non-Hirsch ideals accessible by RL, it struggles to scale to larger degrees. In particular, we observe in Fig. 5 that while the average priority score is initially monotonically increasing, it stops once the buffer consists of graphs which satisfy the diameter constraint but have a single irreducible edge. This indicates that for larger degrees, the heuristic alone does not provide sufficiently even-spaced goals to allow the model to reduce the last remaining edge while still satisfying the diameter constraint.

**4. Hindsight Experience Replay ([Andrychowicz et al., 2017](#))**: Given a state $s$, we construct a goal $g_s$ associated to the state $s$ as $g_s = (\mathrm{diam}(s), h(s))$ where $h$ is the heuristic (3). The target goal is $g_{\mathsf{tgt}} = (d + 1, \ 0)$. We shape the goal-based reward function as:

$$r_g(s, a, s') = -[g_{s'} \neq g] . \qquad (11)$$

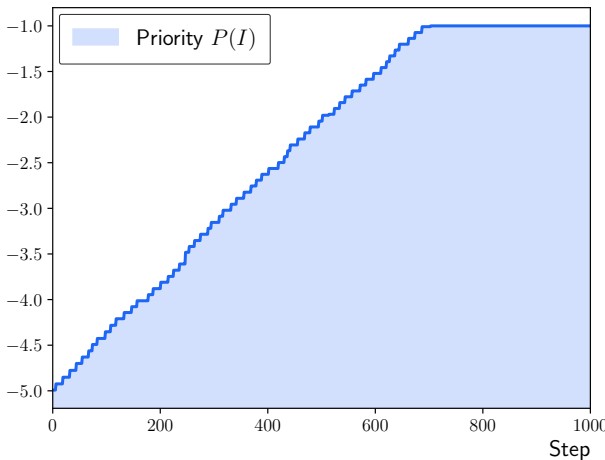

*Figure 5.* Mean priority $P(I)$ of ideals $I$ in the buffer during training for degree $d = 6$.

We observe some improvement for constructing ideals in degree $d = 4$ (See SAC+HER in Table 1), however, the approach struggles to construct ideals in higher degrees.

## 6.2. Chained Constrained Options

Every non-Hirsch ideal contains a spine. Therefore, given that the agent defined in 6.1 builds a non-Hirsch ideal from spines, we may ask whether, given a spine, there always exists a non-Hirsch ideal $I$ for which the spine is a diameter-realizing path for (1). If this holds, we call the spine a *supporting* spine.

For any supporting spine $s_0$, we can construct a trajectory $\tau = (s_0, \dots, s_T)$ where $s_T$ is a non-Hirsch ideal, such that every intermediate state $s_t$ satisfies the diameter constraint $\mathrm{diam}(s_t) > d$ (See Thm. A.8).

By measuring the likelihood of randomly sampled spine being a supporting spine in the sense defined above, we find that $\ll 1\%$ of spines have this property (see Best-first search in Table 1). This observation motivates us to reformulate the reinforcement learning framework within constrained options framework with two chained temporal abstractions: spines (S) and linearizations (L). We illustrate this procedure in Fig. 2. In particular, we extend the notion of options by incorporating intra-option policy constraints. More specifically, we introduce options:

$$\omega_S = (I_S, \pi_S, \mathcal{C}_S, \beta_S), \quad \omega_L = (I_L, \pi_L, \mathcal{C}_L, \beta_L), \quad (12)$$

where $\mathcal{C}_S$ and $\mathcal{C}_L$ are constraints on the intra-option policies $\pi_S$ and $\pi_L$, respectively. We define the initialization set $I_S$ of the spine policy to consist of single-node graphs and $\beta_S = I_L$ as the set of all path graphs of diameter strictly larger than the degree. The constraint $\mathcal{C}_S$ for procedurally

constructing a supporting spine is essentially restricting the intra-option policy $\pi_S$ to ensure monotonicity of the diameter:

$$\mathcal{C}_S(s_t) = \{a_t \in \mathcal{A} \mid \mathrm{diam}(s_t) < \mathrm{diam}(a_t(s_t)) < \infty\}, \quad (13)$$

whereas, for the linearization option $\omega_L$, the constraint $\mathcal{C}_L$ ensures that the diameter of the subsequent states remain above the degree:

$$\mathcal{C}_L(s_t) = \{a_t \in \mathcal{A} \mid \mathrm{diam}(a_t(s_t)) > d\}. \quad (14)$$

Due to a specific choice of the terminations $\beta_S$ and $\beta_L$, the distribution $p(\tau)$ of the trajectories (Bacon et al., 2017) sampled using options $\omega_S$ and $\omega_L$ takes a particularly simple form:

$$p(\tau) = p(s_0) \prod_{t'=T_S}^{T_S+T_L-1} p(s_{t'+1}|s_{t'}, a_{t'})\pi_L(a_{t'}|s_{t'}) \cdot \quad (15)$$
$$\prod_{t=0}^{T_S-1} p(s_{t+1}|s_t, a_t)\pi_S(a_t|s_t),$$

where $T_S$ denotes the number of steps required for an agent to construct a spine of diameter $> d$ starting from a single node. Generally, for stability purposes, we choose the threshold on the diameter to be slightly above $d$ and allow the agent to execute the linearization option $\omega_L$ at each step satisfying (1).

We establish the baseline by letting $\pi_S^{\mathrm{baseline}}(a_t|s_t) = \mathcal{U}(\mathcal{C}_S(s_t))$, the uniform distribution on $\mathcal{C}_S(s_t)$, and $\pi_L^{\mathrm{baseline}}(a_t|s_t)$ be restricted A* search policy $\pi_L^{A^*,h}(a_t|s_t)$ using heuristic $h(I)$ from (3) with intra-option constraint $\mathcal{C}_L$ from (14). While likelihood of randomly constructing a supporting spine via a baseline policy $\pi_S^{\mathrm{baseline}}$ is small, the linearization policy $\pi_L^{A^*,h}$ using heuristic $h$ from (3) successfully terminates in all cases that we have tested. This is mostly due to the fact that heuristic seems to provide sufficient information for introducing new nodes which eliminate irreducible edges. Furthermore, we observe that the mean number of steps required to linearize a supporting spine using the policy $\pi_L^{A^*,h}$ scales linearly with the degree $d$ (see Fig. 8). Therefore, we shall fix $\pi_L = \pi_L^{\mathrm{baseline}}$ and focus on optimizing the spine policy $\pi_S$. For assigning the rewards, we use goal-hitting sparse-reward function:

$$r(s, a, s') = \begin{cases} 1 & \text{if } s' \text{ is non-Hirsch} \\ 0 & \text{otherwise} \end{cases}. \quad (16)$$

While PPOC (Klissarov et al., 2017) extends PPO to optimize the terminations, given that we do not parametrize terminations, we focus only on update rules corresponding to the policies.

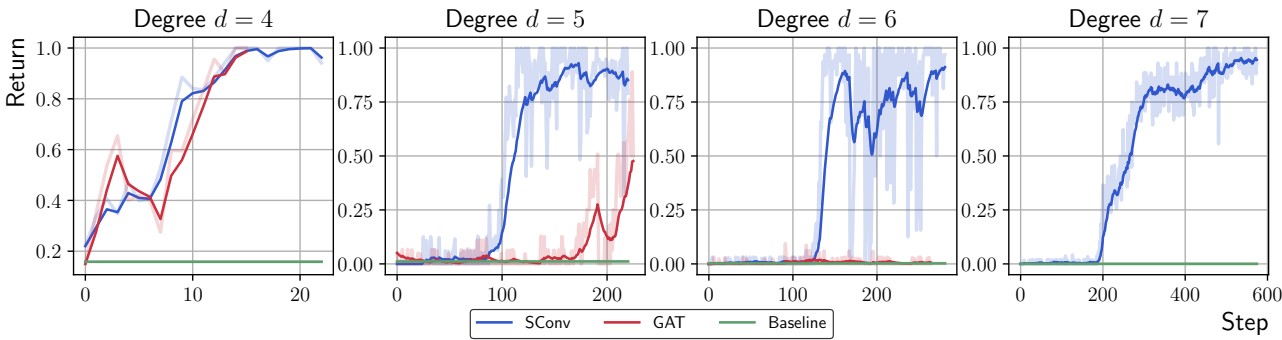

*Figure 6.* Average episodic return over degrees $d = 4$ to $d = 7$. The SConv (Section 4) consistently outperforms the baseline and GAT models. Both SConv and GAT were trained by optimizing $\pi_S$ intra-option hard-constrained policy. We used $1/10$ the learning rate for $d = 7$ to mitigate stability issues. The inflection points of GAT for degrees $d > 5$ occur far beyond the number of steps displayed in the figures and are therefore omitted.

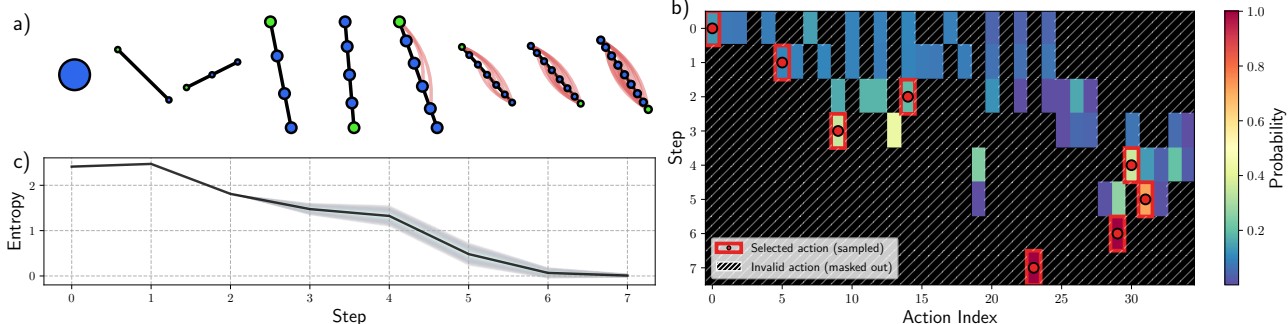

*Figure 7.* (a) Visualization of a trajectory sampled using learned intra-option spine policy for $d = 4$, with the policy shown in (b). (c) shows the entropy of the admissible actions is decreasing as the number of cycle-free actions decreases with the number of nodes.

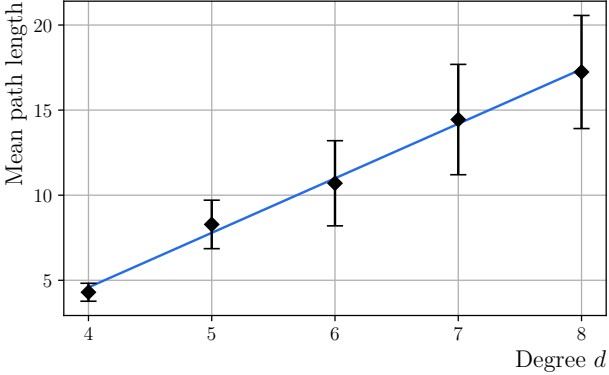

*Figure 8.* Mean path length of the heuristic $h$-based linearization policy $\pi_L^{A^*,h}$ constructing non-Hirsch ideal from a supporting spine.

A major difficulty is correctly constraining the policies without sacrificing the stability of the training. This leads us to consider two different strategies for enforcing the constraints $\mathcal{C}_S$ and $\mathcal{C}_L$: *hard constraints* and *soft constraints*, which we discuss in depth below.

**Hard Constraints (Huang & Ontañón, 2022):** Given a state $s_t$ at step $t > 0$, we define a mask which replaces the logits of the actions $a_t \notin \mathcal{C}_S(s_t)$ by a large negative number $M \approx -\infty$. This effectively restricts the allowed set

of actions whilst maintaining well-define policy-gradients (Huang & Ontañón, 2022). However, the key difficulty is the computational complexity of verifying set membership $a_t \overset{?}{\in} \mathcal{C}_S(s_t)$ due to the combinatorial scaling (2) of the number of actions $\mathcal{A}$. Since this is unavoidable, we address this by parallelizing the membership checks over multiple cores.

**Soft Constraints (Ray et al., 2019):** An alternative approach is to add a adaptive penalty to the agent for violating the constraints. However, we observe that this approach doesn't produce solutions faster than the baseline. See Appendix C for more details. Therefore, we mainly focus on policies with hard constraints.

The average episodic returns of agents trained using the hard constraints are shown in Fig. 6. We observe that despite masking out invalid actions in the policy, the training remains stable. The agents are able to learn useful intra-option policy $\pi_S$, which significantly outperforms the baseline. Furthermore, both the baseline and constrained options policy outperform other RL approaches, as evidenced by the results in Table 1.

In Fig. 6, we also present a brief comparison of the Syzygy-Aware graph convolutions (Section 4) versus regular Graph

Attentions. We observe that SConv exhibits earlier convergence compared to Graph Attention, which we attribute to the fact that the attention matrices between pairs of nodes resemble the bit-wise binary operations required in syzygy reduction algorithm 1.

Visualizing the actions performed by the agent in Fig. 7, we observe that the constraints significantly reduce the space of valid actions. Furthermore, as the agent builds a spine, this space continues to shrink, as indicated by decreasing entropy. We list some of the examples of non-Hirsch ideals of degree $d = 7$ constructed using RL agent in the Appendix B.

## 7. Conclusion

In this work, we demonstrate that temporal abstractions are not only beneficial but also crucial for solving an extremely sparse-rewards search problem in commutative algebra. By analyzing the failure modes of the standard end-to-end RL algorithms, we were able to identify rare bottleneck states, called supporting spines, which have allowed us to reformulate the problem within the framework of constrained options in HRL.

Our results show that while the linearization of a supporting spine can be easily accomplished via available heuristics, the key difficulty lies in discovering such spines in the first place. Learning a dedicated policy for constructing such spines is therefore the primary objective and leads to substantial gains compared to what was previously accessible. To our knowledge, this is the first demonstration that learned temporal abstractions enable sparse-reward search in commutative algebra and, more broadly, in similar environments where solutions are vanishingly rare.

Beyond this specific application, our work motivates a general paradigm for approaching "needle-in-a-haystack" agentic search problems in mathematics: identify the bottleneck states by observing the behavior of standard RL algorithms and design temporal abstractions which significantly reduce the complexity of the problem.

## Acknowledgements

The project was sponsored by the Defense Advanced Research Projects Agency under cooperative agreement HR0011262E017, by the NSF AIMing grant 2522494, by the DRW Foundation, and by a philanthropic gift from Les Kohn. M.T. is also supported by the U.S. Department of Energy (Grant No. DE-SC0011632) and by the Walter Burke Institute for Theoretical Physics. Additionally, this work was supported with Cloud TPUs from Google's TPU Research Cloud (TRC), with GPUs from the NVIDIA Academic Grant Program, by cloud computing resources provided by Nebius through the Research Program of Nebius Academy, and by Advanced Micro Devices, Inc. under the AMD University Program's AI & HPC Cluster. The content of the information does not necessarily reflect the position or the policy of the Government, and no official endorsement should be inferred.

**Approved for public release; distribution is unlimited.**

## Impact Statement

This paper presents work whose goal is to advance the field of machine learning and mathematical reasoning. There are many potential societal consequences of our work, none of which we feel must be specifically highlighted here.

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

# A. Mathematical Problem: Relevant Background

Monomial ideals serve as a bridge between commutative algebra and combinatorics. A direct connection between the Hirsch conjecture and the linear presentation of monomial ideals is implicated in a conjecture proposed by Gil Kalai in an algebraic form (Kalai, 2009): Consider the collection $\mathcal{G}'(k, n)$ of graphs $G$ whose vertices are labeled by $k$-subsets of an $n$ element set, in such a way that if $v$ is a vertex labeled by the set $S$ and $u$ is a vertex labeled by the set $T$, then there is a path between $u$ and $v$ so that all labels of its vertices are sets containing $S \cap T$.

**Conjecture A.1.** *Kalai's Abstract Polynomial Hirsch Conjecture (APHC): If $G \in \mathcal{G}(k, n)$ is connected, then the diameter of $G$ is bounded above by a polynomial in $k$ and $n$.*

The original Hirsch conjecture is related to this as follows. Let $P$ be a polytope in $N$-space, with facets labeled $1 \ldots, n$, and let each vertex be labeled by the sets of $N$ facets that intersect in it (thus a vertex may have several labels). If $v$ is a vertex labeled by a set $S$ and $u$ is a vertex labeled by a set $T$ and $S \cap T$ is non-empty, then $u$ and $v$ lie on the same facet and are obviously connected by a path traversing the edges of this facet. Thus, Kalai's conjecture would say that the diameter of the 1-skeleton of $P$ is bounded by a polynomial in $n$ and $k$. Of course, the graph formed by the skeletons of polytopes is very special.

Let $d = n - k$, and consider a relabeling of the vertices of graphs in $\mathcal{G}'(k, n)$ with the complements of the original labels. Thus, we arrive at the set $\mathcal{G}(d, n)$ of connected graphs whose vertices are labeled by $d$-element subsets of $1, \ldots, n$ in such a way that *If there is a path between vertices $u$ and $v$, then there is a path consisting of vertices whose labels are subsets of $A \cup B$.*

**Proposition A.2.** *(Dao & Eisenbud, 2022) Let $G$ be a graph whose vertices are labeled by $d$-element subsets of $1, \ldots, n$ as above, and let $I$ be the (square-free) monomial ideal in a polynomial ring $k[x_1, \ldots, x_n]$ generated by monomials*

$$\{x_{i_1} \cdots x_{i_d} \mid \{i_1, \ldots, i_d\} \text{ is the label of a vertex of } G\}$$

*corresponding to the vertex labels. The graph $G$ belongs to $\mathcal{G}(d, n)$ if and only if the ideal $I$ has a linear presentation.*

Let $I$ be the (square-free) monomial ideal in a polynomial ring $k[x_1, \ldots, x_n]$ generated by monomials corresponding to the vertex labels,

$$\{x_{i_1} \cdots x_{i_d} \mid \{i_1, \ldots, i_d\} \text{ is the label of a vertex of } G\}$$

Given any square-free monomial ideal $I \subset k[x_1, \ldots, x_n]$ we form a labeled graph $G_I$ whose vertices are the minimal monomial generators, and whose labels are the indices of the factors of the generators. With this language we may restate Kalai's conjecture as:

**Conjecture A.3.** *If $I$ is a square-free monomial ideal in $n$ variables, generated in degree $d$ and having a linear presentation, then the diameter of $G_I$ is bounded above by a polynomial in $d$.*

*Remark* A.4. The process of *polarization* can be used to replace any monomial ideal by a square-free one with a similar presentation, so the conjecture is equivalent to the same statement for all monomial ideals, taking the labels as multisets or as the monomials themselves.

For very small $d$, the Conjecture A.3 holds true and even satisfies the original Hirsch-style bound $\mathrm{diam}(I) \leq d$, *cf.* (1). For example, with $d = 2$, consider any two vertices $ab, cd$. By local connectedness, there is a path between them involving only variables $a, b, c, d$, which is easily seen to be, up to renaming, $ab, bc, cd$. For $d = 3$, see (Morales et al., 2014). More generally, for each $d$, there is a finite list of "obstructions" to linearity. But these lists grow rapidly with $d$ and so far have not been very useful for $d > 3$.

A central problem in addressing Conjecture A.3, and more generally in studying the linear presentation of monomial ideals, is to find examples of monomial ideals $I$ generated in degree $d$ having linear presentation where the diameter of $G_I$, divided by $d$, is as large as possible. So far, the best we can do "by hand" is an ideal in 10 variables $a, \ldots j$ for which $d = 7$ and the diameter of $G_I$ is 9, shown in Fig. 9.

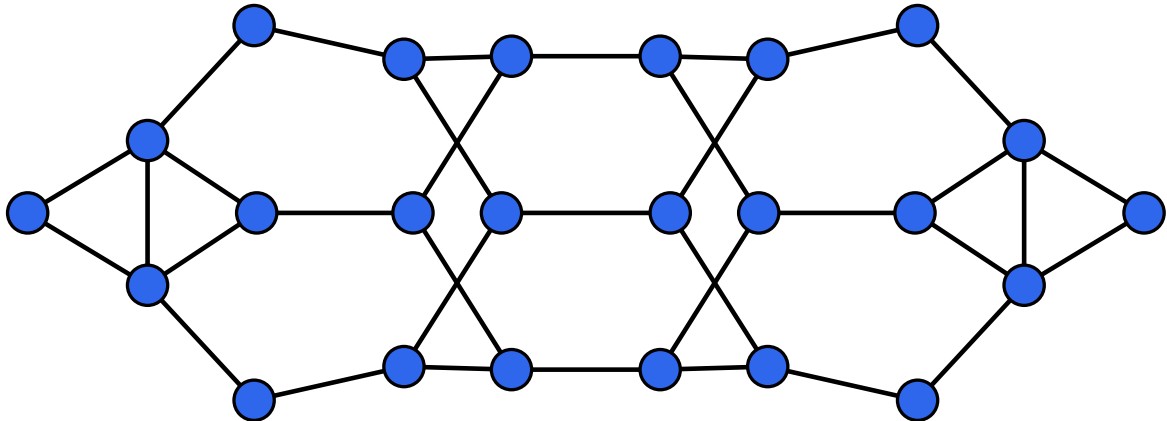

defghij,befghij,aefghij,acefgij,acdfgij,abdfgij,abdegij,bcdegij,
bcdefij,acdefhj,abefghj,abdfghj,bcdfghj,acdeghj,bcdeghj,abcdefj,
bcefghi,bcdfghi,acdfghi,acdeghi,abdeghi,abdefhi,abcdefi,abcdefg

*Figure 9.* A non-Hirsch ideal of degree $d = 7$ for which the diameter is 9.

## A.1. Edge Reduction Algorithm

Let $R = k[x_1, \ldots x_n]$ be a polynomial ring. By applying Hilbert Syzygy Theorem to $I$ treated as a finitely generated $R$-module, we may construct its minimal finite resolution:

$$0 \longrightarrow F_n \longrightarrow \cdots \longrightarrow F_1 \xrightarrow{\phi_1} F_0 \xrightarrow{\phi_0} I \longrightarrow 0 \, , \tag{17}$$

where each $R$-module $F_i$ for $0 \leq i \leq n$ is finitely generated and free. Each $F_{0 \leq i \leq n}$ may further be decomposed into a direct sum of degree-shifted modules:

$$F_i \simeq \bigoplus_{j=0}^{m_i} R(-i-j)^{\beta_{ij}} \, , \tag{18}$$

where $\beta_{ij} = \dim_k \operatorname{Tor}_i^R(I, k)_j$ are the Betti numbers and $\sum_{j=0}^{m_i} \beta_{ij} = \operatorname{rank} F_i$. A monomial ideal $I$ is *linear* if all maps in (17) are of degree one. We are interested in a slightly weaker condition, where we are mainly focused on the linearity of the presentation of the ideal $I$ in the resolution. In particular, we define $I$ to be *linearly presented* if the map $\phi_1$ in the resolution (17) has degree one. We may check if an ideal is linearly presented by computing the reduced Gröbner basis of the first syzygy module $\operatorname{Syz}_1(I)$, defined as $\ker \phi_0$. While generally this can be accomplished using standard computer algebra frameworks, such as Macaulay2 (Grayson & Stillman), we observe that these methods are typically infeasible for ideals generated by a large number of monomials. This is especially important for reinforcement learning applications, where any computational cost during environment interactions may lead to prohibitively slow training as each action requires us to perform a linear presentation test. In order to mitigate the computational costs associated to the environment, we propose an alternative parallelized algorithm which allows us to utilize the computational advantages offered by modern CUDA-enabled GPUs.

Let $I = (f_i)_{i=1}^g$ where each $f_i$ is a monomial in variables $\{x_1, \ldots, x_n\}$, such that no two generators are the same and $\deg f_i = d$. Note that the set of monomials $\{f_i\}_{i=1}^g$ is a Gröbner basis of $I$ as by construction $\operatorname{in}_> f_i = f_i$, thus Buchberger's criterion is trivially satisfied. Furthermore, by applying Schreyer's theorem (Eisenbud, 1995) to $I$, it is easy to see that an unreduced Gröbner basis of $\operatorname{Syz}_1(I)$ is given by:

$$S_{ij} := \frac{\operatorname{lcm}(f_i, f_j)}{f_j} \epsilon_j - \frac{\operatorname{lcm}(f_i, f_j)}{f_i} \epsilon_i = m_{ij} \epsilon_j - m_{ji} \epsilon_i \, , \tag{19}$$

where $\{\epsilon_i\}_{i=1}^g$ is the canonical basis of $F_0$ such that $\phi(\epsilon_i) = f_i$. Unfortunately, such basis is typically far from being minimal. Checking if an ideal $I$ is linearly presented amounts to checking if the generators (19) of degree $> d + 1$ can be expressed in terms of generators of smaller degree. While general heuristics, such as Gebauer–Möller conditions (Gebauer

---

**Algorithm 1** Edge Reduction Algorithm

---

1: **Input:** edge $e = (v, v') \in E_2$. *// Or any $E_{l>1}$*
2: **Input:** first-level graph $G_1 = (V, E_1)$.
3: **Output:** True if $e$ is reducible. False otherwise.
4: ────────────────
5: **Define**: $m_{ww'} := w' \,\&\, \neg w$ for all $w, w' \in V$.
6: **Define**: $\mathcal{N}_v(G_1) := $ 1-hop neighborhood of $v \in V$ in $G_1$.
7: $\mathcal{V} \leftarrow \emptyset$   *// visited nodes*
8: Initialize empty queue $q$
9: Enqueue $q \leftarrow (v, m_{vv'})$
10: **while** $q$ is not empty **do**
11:    Dequeue $(w, m) \leftarrow q.\text{pop}()$
12:    **for** each $w' \in \mathcal{N}_w(G_1)$ **do**
13:      **if** $w' \in \mathcal{V}$ **then**
13:        **continue**
14:      **end if**
15:      **if** $m \mid m_{vw'} = m$ **then**
16:        **if** $w' = v'$ **then**
16:          **return** True   *// The edge $e$ was successfully reduced*
17:        **end if**
18:        $r \leftarrow m \,\&\, \neg m_{ww'}$   *// Missing variables coming from $w$*
19:        $q \leftarrow (w', \, r \mid m_{w'w})$   *// Append missing variables to $m_{w'w}$*
20:      **end if**
21:    **end for**
22:    $\mathcal{V} \leftarrow \mathcal{V} \cup \{w\}$
23: **end while**
24: **return** False   *// If no path was found, then edge is irreducible*

---

& Möller, 1988) exist, those typically do not yield minimal bases. Furthermore, we are interested in refining the obstruction to linearity, in order to define a heuristic measuring how far away a given ideal is from being linearly presented. Therefore, we focus on the following objectives:

**1. Features**: Using the data associated to the first syzygy module $\text{Syz}_1(I)$, provide an additional set of features which might be useful for the agent to identify useful moves and measure how far ideal $I$ is from being linearly presented.

**2. Parallelization**: Design a GPU-accelerated algorithm which efficiently checks if a given ideal $I$ is linearly presented.

Inspired by the results of (Dao & Eisenbud, 2022), we design a reduction algorithm which reformulates the problem as graph reduction algorithms and provides a new set of features which we use to augment the graph $G_I$.

### A.1.1. COMBINATORIAL DATA OF THE FIRST SYZYGY MODULE

Given a square-free monomial ideal $I = (f_i)_{i=1}^g$ of degree $d > 0$, we associate a multi-level graph $G_{1 \le l \le d} = (V, E_l)$ to the first syzygy module $\text{Syz}_1(I)$ of the ideal $I$. We define the vertices $V$ of $G_l$ as $\{v_1, \ldots, v_g\}$, where $g$ is the number of generators of $I$. Furthermore, we associate each generator $f_i$ with $v_i$ via a map $\theta(v_i) = f_i$. For each level $l > 0$, we define the set of edges $E_l$ as:

$$E_l := \{(v, w) \in V \mid \deg m\,(\theta(v), \theta(w)) = l\}, \tag{20}$$

where $m(f, g) := {}^{\text{lcm}(f,g)}\!/_g$. Note that the edges $E_l$ are in one-to-one correspondence with the generators (19):

$$E_l \ni (\theta^{-1}(f_i), \theta^{-1}(f_j)) \overset{s}{\longmapsto} S_{ij}, \tag{21}$$

therefore, all linear generators of $\text{Syz}_1(I)$ correspond to the edges $E_1$ of the first-level component $G_1$ of $G$. Furthermore, testing if the ideal $I$ is linearly presented amounts to checking the properties of the graphs $G_{l>1}$. In particular, we define:

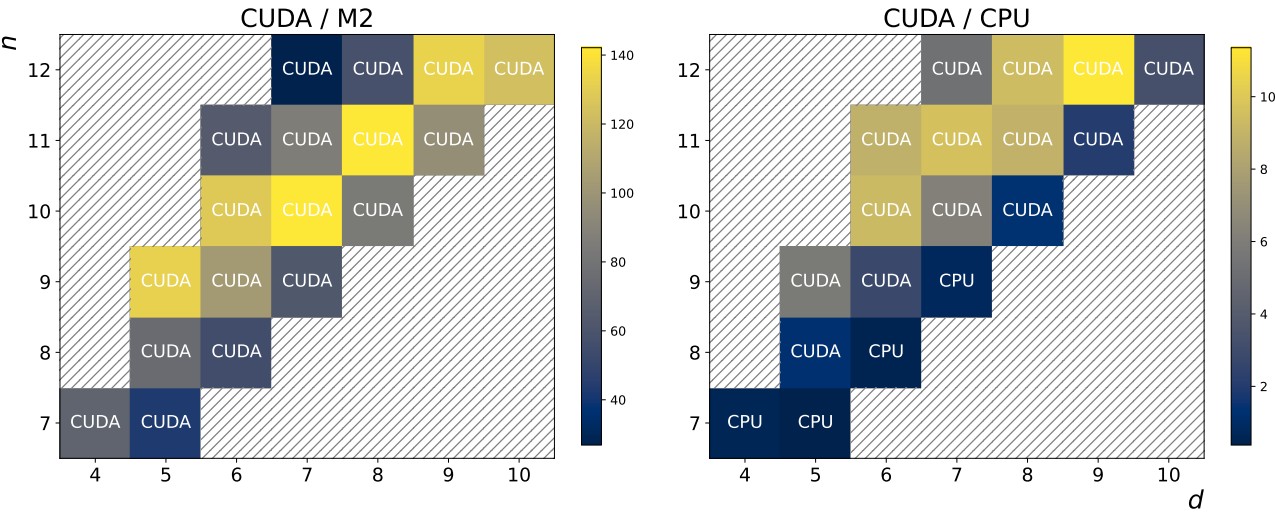

*Figure 10.* End-to-end timing benchmarks of linear presentation tests across different platforms.

**Definition A.5.** An edge $e \in E_{l>1}$ is *reducible* if the generator $s(e) \in \mathrm{Syz}_1(I)$ can be written as an $R$-linear combination:

$$s(e) = \sum_{e' \in E_1} r(e')s(e'), \tag{22}$$

for some $r \colon E_1 \to R$. Otherwise, we call $e$ *irreducible*.

From this definition, we immediately obtain a characterization of linearly presented ideals in terms of reducible edges:

**Theorem A.6.** *An ideal $I$ is linearly presented iff every edge $e \in E_{l>1}$ is reducible.*

*Proof.* Let $I$ be linearly presented, then $\mathrm{Syz}_1(I)$ is generated by linear forms, corresponding to the edges in $E_1$. Since the generators associated to all edges $\cup_l E_l$ form a Gröbner basis of $I$, we see that the generators associated to the edges $\bigcup_{l>1} E_l$ must be expressible in terms of generators $s(E_1)$ via (22), thus making them reducible. Conversely, if every edge is reducible, then the generators associated to them may be expressed in terms of generators associated to the edges in $E_1$, thereby making $s(E_1)$ a linear basis of $I$. $\square$

A major consequence of the Theorem A.6 is that we may parallelize the test of whether the ideal is linearly presented over the edges in $\bigcup_{l>1} E_l$. This is possible due to the fact that reducibility of each edge is independent from another. Additionally, the number of irreducible edges measures the extent to which a given ideal fails to be linearly presented, which we use to define heuristic (3).

### A.1.2. ALGORITHM FOR TESTING REDUCIBILITY OF EDGES

Noting that the ideals we consider are square-free, we may encode each generator as a binary sequence of length $n$, where bit at location $i$ is set to 1 if the generator contains variable $x_i$. We denote this encoding by $b \colon \{f_i\}_i \to \mathbb{N}$. This is equivalent to the construction described in (4). Using this representation, the operations $\mathrm{lcm}$ and $\gcd$ over the generators $f_j$ of $I$ reduce to bitwise operations over the binary sequences given by:

$$\mathrm{lcm}(f, g) = b(f) \mid b(g) \tag{23}$$
$$\gcd(f, g) = b(f) \,\&\, b(g)$$

for $f, g$ generators of $I$. Note that each cycle in the graph $G$, consisting of edges $E' \subset \bigcup_l E_l$ corresponds to a relation of form:

$$\sum_{e \in E'} r(e)s(e') = 0, \tag{24}$$

for some $r\colon \bigcup_l E_l \to R$ nonzero no $E'$ (we take $r(e) = 0$ for $e \notin E'$). Since for any given edge $e \in E_{l>1}$, we are interested in finding a relation of form (22), we design an algorithm which verifies if there exists a cycle consisting of edges $E_1 \cup \{e\}$ such that the corresponding relation (24) has coefficient $r(e) = \pm 1$ for the edge $e$. Without loss of generality, we focus on reducibility of edges in $E_2$, noting that the extension to higher levels $E_{l>2}$ is routine. The algorithm is shown in Alg. 1.

**Theorem A.7.** *An edge $e \in E_{l>1}$ is reducible iff the Edge Reduction Algorithm 1 outputs* True.

*Proof.* The algorithm 1 is a BFS search of a cycle consisting of edges $E_1 \cup \{e\}$ containing $e$ which produces a relation (24) such that $r(e) = 1$. Suppose that $e$ is reducible, then there exists a cycle $(v_0, \ldots, v_{k-1})$ consisting of edges $e_i = (v_i, v_{i+1 \bmod k}) \in E_1 \cup \{e\}$. W.l.o.g. let $e_0 = e$, and let $r\colon \cup_l E_l \to R$ be the coefficient function with $r(e_0) = 1$. Note that apart from $e_0$, the only other edge containing $v_1$ is $e_1$, therefore, in order to cancel out the $v_1$ component of the generator $s(e_0)$, we must have $m_{2,1}$ divide $m_{0,1}$, where for brevity we've used $m_{i,j} := m(\theta(v_i), \theta(v_j))$. We verify this in the $m \mid m_{ww'} == m$ step of the algorithm. The ratio $r(e_1) = m_{0,1}/m_{2,1}$ is then stored in variable $r$ of the algorithm. Given $r(e_i)$, constructing $r(e_{i+1})$ is done similarly by checking if $m_{i+2,i+1}$ divides $r(e_i)m_{i,i+1}$ and storing the ratio in variable $r$, where $i + 1$ and $i + 2$ are mod $k$. Ultimately, the cycle reaches $e_0$, thus successfully terminating the algorithm to true.

Conversely, suppose that the algorithm terminates to true for an edge $e \in E_2$. This can only happen if there exists a cycle connecting $(v_0, v_1, \ldots, v_{k-1})$ the endpoints of an edge $e = (v_0, v_1)$ such that $m_{2,1}$ divides $m_{0,1}$, with ratio being $r(e_1)$. Construct the next coefficients recursively using:

$$r(e_{i+1}) = \frac{r(e_i)m_{i,i+1}}{m_{i+2,i+1}}, \tag{25}$$

all of which exist and are well-defined by construction. It is easy to see that $r(e_i)s(e_i)$ forms a telescoping series, leaving:

$$\sum_{i=0}^{k-1} r(e_i)s(e_i) = -m_{1,0}\epsilon_{v_0} + \cdots + r(e_{k-1})m_{k-1,0}\epsilon_{v_0}. \tag{26}$$

Since by construction $r(e_{k-1})m_{k-1,0}$ divides $m_{1,0}$ and $\deg m_{1,0} = 2$, noting that $\deg r(e_{k-1}) > 0$ (propagated from $\deg r(e_0) = 1$) and $\deg m_{k-1,0} = 1$, we must have $r(e_{k-1})m_{k-1,0} = m_{1,0}$, thus $r(e_k) = 1$. This makes $r$ a well-defined coefficient function, which implies that $e$ is indeed reducible. $\square$

By encoding the graph structure of each level $G_j$ in CSR format and parallizing the algorithm 1 over the edges $\bigcup_{l>1} E_l$, we are able to utilize the full advantage provided by the CUDA-accelerated hardware. The timing benchmarks are provided in Fig. 10. Furthermore, by counting and retaining the list of irreducible edges, we obtain a useful set of additional features which help guide the agent towards non-Hirsch ideals.

## A.2. Supporting Spines

**Theorem A.8.** *Given a supporting spine $s_0$, there exists a trajectory $\tau = (s_0, \ldots, s_T)$ to a non-Hirsch ideal $s_T$ such that every intermediate state $s_t$ has diameter* $\mathrm{diam}(s_t) > d$.

*Proof.* This is because we may construct a trajectory from $s_0$ to $s_T$ by sequentially adding vertices from $s_T$ which are not in $s_t$. The fact that $s_0$ is a diameter-realizing path ensures that no new vertex will introduce a strictly shorter path between the endpoints of $s_0$. This can be seen by contradiction: suppose that $s_t$ does not satisfy the diameter constraint for some $t < T$. This implies that there exists a path in $s_t$ between the endpoints of $s_0$ which is shorter than $d + 1$. However, this path would also be a subgraph of $s_T$, which contradicts the assumption that $s_0$ is diameter-realizing in $s_T$. $\square$

# B. Non-Hirsch Ideals Constructed Using HRL

Here we provide visualizations of some of the degree $d = 7$ non-Hirsch ideals in polynomial ring $k[a, b, c, d, e, f, g, h, i, j]$.

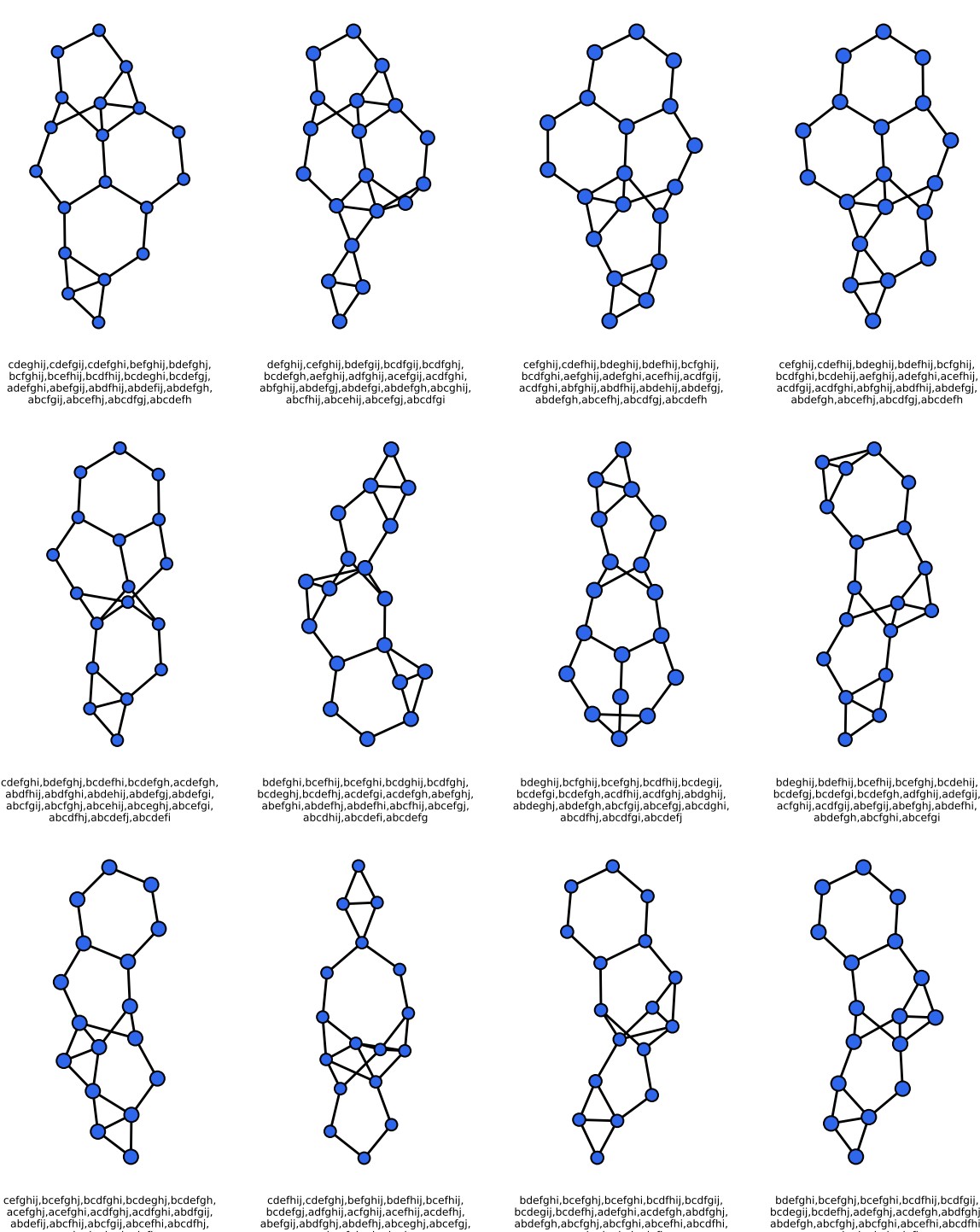

cdeghij,cdefgij,cdefghi,befghij,bdefghj,
bcfghij,bcefhij,bcdfhij,bcdeghi,bcdefgj,
adefghj,abefghj,abdfhij,abdefij,abdefgh,
abcfgij,abcefhj,abcdfgj,abcdefh

defghij,cefghij,bdefgij,bcdfgij,bcdfghj,
bcdefgh,aefghij,adfghij,acefgij,acdfghi,
abfgij,abdefgj,abdefgi,abdefgh,abcghij,
abcfhij,abcehij,abcefgj,abcdfgi

cefghij,cdefhij,bdeghij,bdefhij,bcfghij,
bcdfghi,aefghij,adefghi,acefhij,acdfgij,
acdfghi,abfghij,abdfhij,abdehij,abdefgj,
abdefgh,abcefhj,abcdfgj,abcdefh

cefghij,cdefhij,bdeghij,bdefhij,bcfghij,
bcdfghi,bcdehij,aefghij,adefghi,acefhij,
acdfgij,acdfghi,abfghij,abdfhij,abdefgj,
abdefgh,abcefhj,abcdfgj,abcdefh

cdefghi,bdefghj,bcdefhi,bcdefgh,acdefgh,
abdfhij,abdfghi,abdehij,abdefgj,abdefgi,
abcfghj,abcfghi,abcehij,abceghj,abcefgi,
abcdfhj,abcdefj,abcdefi

bdefghi,bcefhij,bcefghi,bcdghij,bcdfghj,
bcdeghj,bcdefhj,acdefgi,acdefgh,abefghj,
abefghi,abdefhj,abdefhi,abcfhij,abcefgj,
abcdhij,abcdefi,abcdefg

bdeghij,bcfghij,bcefghj,bcdfhij,bcdegij,
bcdefgj,bcdefgh,acdfhij,acdfghj,abdghij,
abdeghj,abdefgh,abcfgij,abcefgj,abcdghi,
abcdfhj,abcdfgi,abcdefj

bdeghij,bdefhij,bcefhij,bcefghj,bcdehij,
bcdefgj,bcdefgi,bcdefgh,adfghij,adefgij,
acfghij,acdfgij,abefgij,abefghj,abdefhi,
abdefgh,abcfghi,abcefgi

cefghij,bcefghj,bcdfghi,bcdeghj,bcdefgh,
acefghj,acefghi,acdfghj,acdfghi,abdfgij,
abdefij,abcfhij,abcfgij,abcefhi,abcdfhj,
abcdegh,abcdefj

cdefhij,cdefghj,befghij,bdefhij,bcefhij,
bcdefgj,adfghij,acfghij,acefhij,acdefhj,
abefgij,abdfghj,abdefhj,abceghj,abcefgj,
abcefgh,abcdegh

bdefghi,bcefghj,bcefghi,bcdfhij,bcdfgij,
bcdegij,bcdefhj,adefghi,acdefgh,abdfghj,
abdefgh,abcfghj,abcfghi,abcefhi,abcdfhi,
abcdegj,abcdefj

bdefghi,bcefghj,bcefghi,bcdfhij,bcdfgij,
bcdegij,bcdefhj,adefghi,acdefgh,abdfghj,
abdefgh,abcfghj,abcfghi,abcefhi,abcdfhi,
abcdegj,abcdefj

*Figure 11.* Some of the $d = 7$ non-Hirsch ideals generated using HRL approach described in Sec. 6.2.

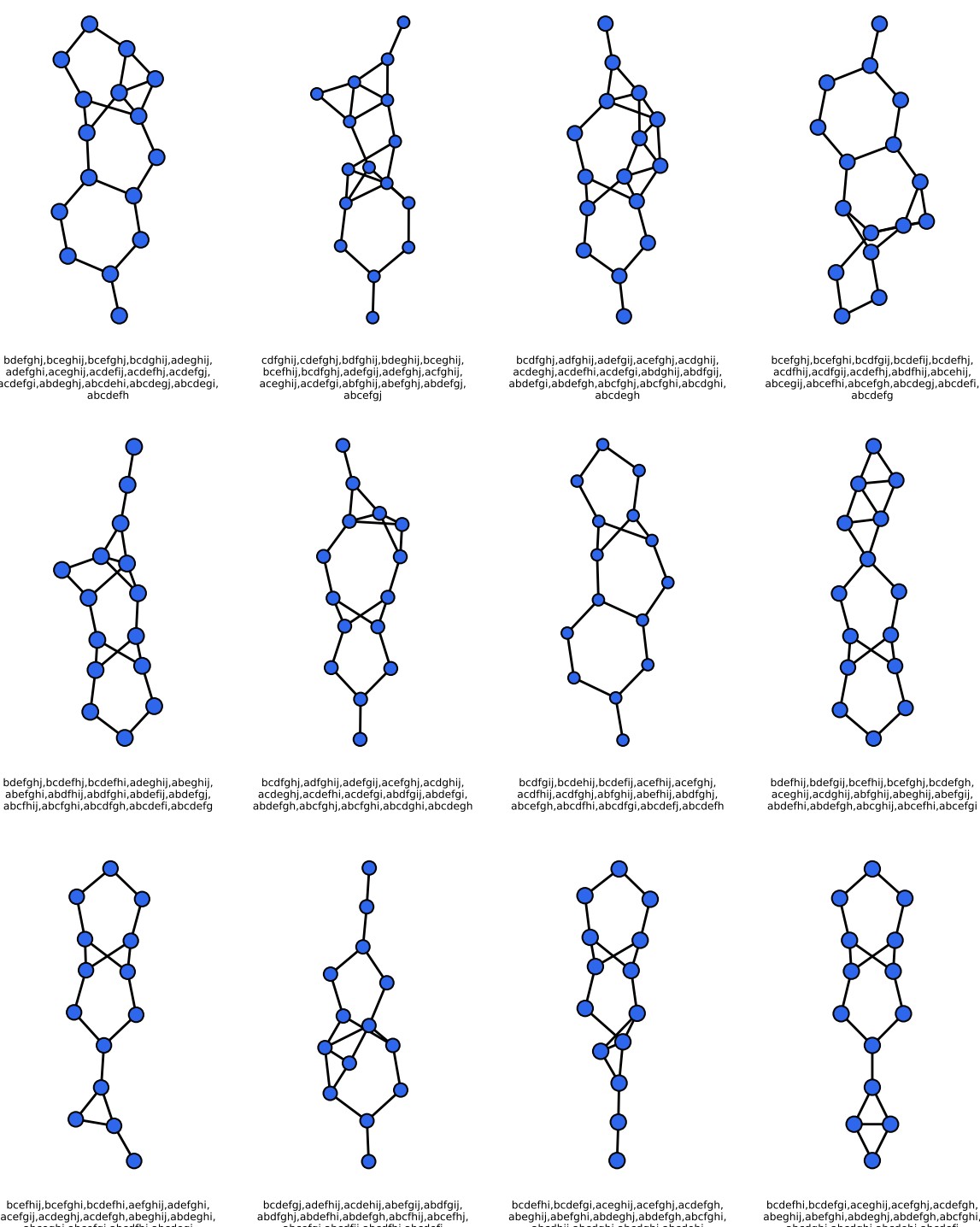

bdefghj,bceghij,bcefghj,bcdghij,adeghij,
adefghi,aceghij,acdefij,acdefhj,acdefgj,
acdefgi,abdeghj,abcdehi,abcdegj,abcdegi,
abcdefh

cdfghij,cdefghj,bdfghij,bdeghij,bceghij,
bcefhij,bcdfghj,adefgij,adefghj,acfghij,
aceghij,acdefgi,abfghij,abefghj,abdefgj,
abcefgj

bcdfghj,adfghij,adefgij,acefghj,acdghij,
acdeghj,acdefhi,acdefgi,abdghij,abdfgij,
abdefgi,abdefgh,abcfghj,abcfghi,abcdghi,
abcdegh

bcefghj,bcefghi,bcdfgij,bcdefij,bcdefhj,
acdfhij,acdfgij,acdefhj,abdfhij,abcehij,
abcegij,abcefhi,abcefgh,abcdegj,abcdefi,
abcdefg

bdefghj,bcdefhj,bcdefhi,adeghij,abeghij,
abefghi,abdfhij,abdfghi,abdefij,abdefgj,
abcfhij,abcfghi,abcdfgh,abcdefi,abcdefg

bcdfghj,adfghij,adefgij,acefghj,acdghij,
acdeghj,acdefhi,acdefgi,abdfgij,abdefgi,
abdefgh,abcfghj,abcfghi,abcdghi,abcdegh

bcdfgij,bcdehij,bcdefij,acefhij,acefghj,
acdfhij,acdfghj,abfghij,abefhij,abdfghj,
abcefgh,abcdfhi,abcdfgi,abcdefj,abcdefh

bdefhij,bdefgij,bcefhij,bcefghj,bcdefgh,
aceghij,acdghij,abfghij,abeghij,abefgij,
abdefhi,abdefgh,abcghij,abcefhi,abcefgi

bcefhij,bcefghi,bcdefhi,aefghij,adefghi,
acefgij,acdeghj,acdefgh,abeghij,abdeghi,
abceghj,abcefgi,abcdfhi,abcdegi

bcdefgj,adefhij,acdehij,abefgij,abdfgij,
abdfghj,abdefhi,abdefgh,abcfhij,abcefhj,
abcefgj,abcdfij,abcdfhj,abcdefi

bcdefhi,bcdefgi,aceghij,acefghj,acdefgh,
abeghij,abefghi,abdeghj,abdefgh,abcfghi,
abcdhij,abcdghj,abcdghi,abcdehi

bcdefhi,bcdefgi,aceghij,acefghj,acdefgh,
abeghij,abefghi,abdeghj,abdefgh,abcfghi,
abcdghj,abcdghi,abcdehi,abcdefi

*Figure 12.* Some of the $d = 7$ non-Hirsch ideals generated using HRL approach described in Sec. 6.2.

In order to verify that these ideals are indeed linearly presented, use the following Macaulay2 script:

```
R = ZZ/101[a,b,c,d,e,f,g,h,i,j];
isLinear = I -> (
    d := (degree I_*_0)_0;
    {d+1} == max degrees source syz gens I
);

I = ideal({insert generators here});

isLinear I
```

## C. Bottleneck States Identified Via Soft Constraints

A standard approach is to employ PPO-Lagrangian as a "soft-constraint" alternative to "hard-constraint" in (13) with no additional hyperparameters. The constraint dynamically penalizes transitions that fail to efficiently produce spines. Specifically, we define a constraint cost function $c_S(s_t) = \mathrm{diam}(s_{t-1}) - \mathrm{diam}(s_t)$ if $s_{t-1}$ is not a terminal state, otherwise 0. Interestingly, PPO-Lagrangian enables the agent to construct a much more general class of bottleneck states, such as trees and tadpole-shaped graphs (See Figure 13). However, we observe that while PPO-Lagrangian learns to satisfy the constraints, the rate at which it produces solutions is significantly lower than that achieved with hard constraints, as expected from prior results (Ray et al., 2019).

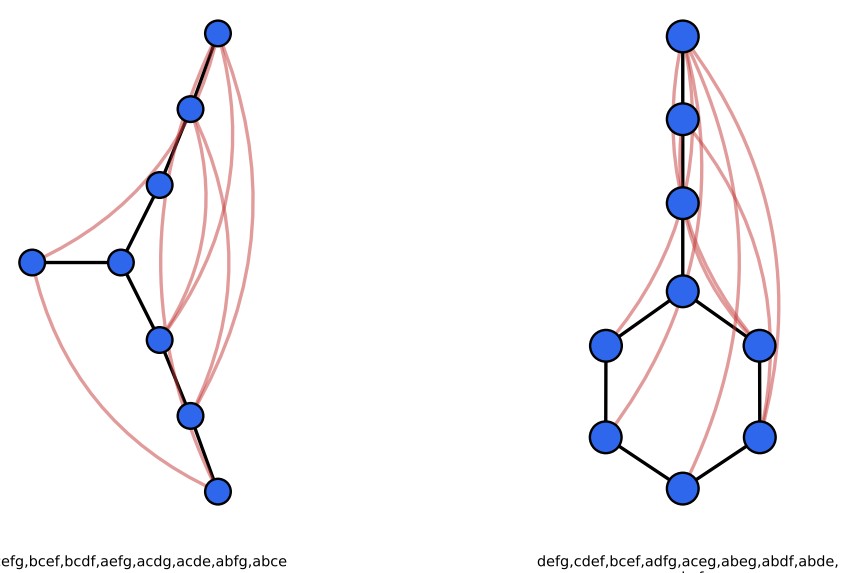

cefg,bcef,bcdf,aefg,acdg,acde,abfg,abce

defg,cdef,bcef,adfg,aceg,abeg,abdf,abde,
abcf

*Figure 13.* Non-spine bottleneck states identified using soft constraints.

# D. Training and Model Details

## D.1. Syzygy-aware Message Passing

---

**Algorithm 2** SConv: Preprocessing

---

**Require:** Binary mask $\mathbf{m}$ for nodes of $G_I$ in $G_{\mathsf{max}}(d, n)$.
**Require:** Node and mask encodings $\phi_{\mathrm{ne}}$ and $\phi_{\mathrm{me}}$, respectively. Embedding dimension $d_e$.
 1: Encode masks:
$$\mathbf{m} \leftarrow \phi_{\mathrm{me}}(\mathbf{m}) \quad \in \mathbb{R}^{|\mathbf{m}| \times d_e}$$

 2: Expand mask features by copying it over $n$ variables:
$$\mathbf{m} \leftarrow \mathrm{Expand}(\mathbf{m}, n) \quad \in \mathbb{R}^{|\mathbf{m}| \times n \times d_e}$$

 3: Encode node and mask features:
$$\tilde{\mathbf{x}} = \phi_{\mathrm{ne}}(\mathbf{x}) + \phi_{\mathrm{me}}(\mathbf{m}) \quad \in \mathbb{R}^{|\mathbf{m}| \times n \times d_e}$$

**return** $\tilde{\mathbf{x}} \in \mathbb{R}^{|\mathbf{m}| \times n \times d_e}$

---

---

**Algorithm 3** SConv: Message Passing

---

**Require:** Base graph $G_{\mathsf{max}}(d, n)$. Binary mask $\mathbf{m}$ for nodes of $G_I$ in $G_{\mathsf{max}}(d, n)$. Irreducible edges $E_{\mathrm{irr}}$.
**Require:** Embedded features $\tilde{\mathbf{x}} \in \mathbb{R}^{|\mathbf{m}| \times n \times d_e}$ from 2
**Require:** Embedding dimension $d_f$. Number of heads $H$.
**Require:** Weights $W_Q, W_K, W_V \in \mathbb{R}^{d_e \times H \times d_f}$.
**Require:** $k$-layer MLP: $f_{\mathrm{MLP}} \colon \mathbb{R}^{n \times H \times d_f} \to \mathbb{R}^{d_f}$.
 1: **for** each edge $e = (i, j) \in E_1(G_{\mathsf{max}}(d, n)) \cup E_{\mathrm{irr}}$ **do**
 2:     Compute key and value:
$$\mathbf{K}_j = \tilde{\mathbf{x}}_j W_K, \quad \mathbf{V}_j = \tilde{\mathbf{x}}_j W_V \quad \in \mathbb{R}^{n \times H \times d_f}$$

 3:     Compute query:
$$\mathbf{Q}_i = \tilde{\mathbf{x}}_i W_Q \quad \in \mathbb{R}^{n \times H \times d_f}$$

 4:     Compute cross-attention:
$$\mathbf{m}_{ij} = \mathsf{Attention}(Q_i, K_j, V_j) \quad \in \mathbb{R}^{n \times H \times d_f}$$

 5:     Apply MLP:
$$\mathbf{m}_{ij} \leftarrow f_{\mathrm{MLP}}(\mathbf{m}_{ij}) \quad \in \mathbb{R}^{d_f}$$

 6: **end for**
 7: Aggregate the features per node:
$$\mathbf{y}_i = \mathrm{MEAN}\{\mathbf{m}_{ij} \mid (i, j) \in E_1(G_{\mathsf{max}}(d, n)) \cup E_{\mathrm{irr}}\} \quad \in \mathbb{R}^{d_f}$$

**return** $\mathbf{y} \in \mathbb{R}^{|\mathbf{m}| \times d_f}$.

---

## D.2. Hyperparameters

| Hyperparameter | Symbol | Value |
|---|---|---|
| Number of heads | $H$ | 4 |
| Node and mask embedding dimension | $d_e$ | 64 |
| SConv embedding dimension | $d_f$ | 64 |

*Table 2.* Model Hyperparameters.

| Hyperparameter | Value | | | |
|---|---|---|---|---|
| | $d = 4$ | $d = 5$ | $d = 6$ | $d = 7$ |
| # parallel environments | 16 | 16 | **32** | **48** |
| # steps per epoch | 128 | 128 | 128 | 128 |
| # improvement epochs | 4 | 4 | 4 | 4 |
| Learning rate | $2.5 \times 10^{-4}$ | $2.5 \times 10^{-4}$ | $2.5 \times 10^{-4}$ | $\mathbf{2.5 \times 10^{-5}}$ |
| Mini-batch size | 64 | 64 | 64 | 64 |
| Max steps for $\pi_S$ | 10 | 11 | 12 | 13 |
| Max steps for $\pi_L$ | 10 | 10 | **15** | **15** |

*Table 3.* Training Hyperparameters.

### D.3. Intrinsic Curiosity Module (Pathak et al., 2017)

Standard RL exploration strategies often fail when extrinsic rewards are extremely sparse or delayed. In these cases, intrinsic reward may offer a solution to the exploration bottleneck. The intrinsic curiosity module of (Pathak et al., 2017) is comprised of three networks: a feature embedding states to features $\phi : s \mapsto \phi(s)$, an inverse model mapping features of consecutive states to a predicted action between them $\rho : (\phi(s), \phi(s')) \mapsto \hat{a}$ and a forward model mapping an action-feature pair to the predicted features of the next state $\sigma : (a, \phi(s)) \mapsto \hat{\phi}(s')$. The prediction error of the forward model is used as the basis for the intrinsic reward signal

$$r^i(s, s') = \eta \left\| \hat{\phi}(s) - \phi(s') \right\|_2^2 \tag{27}$$

where $\eta$ is a scaling factor. The mean-return training curve for an HRL agent, trained using ICM module, is shown in the Figure 14.

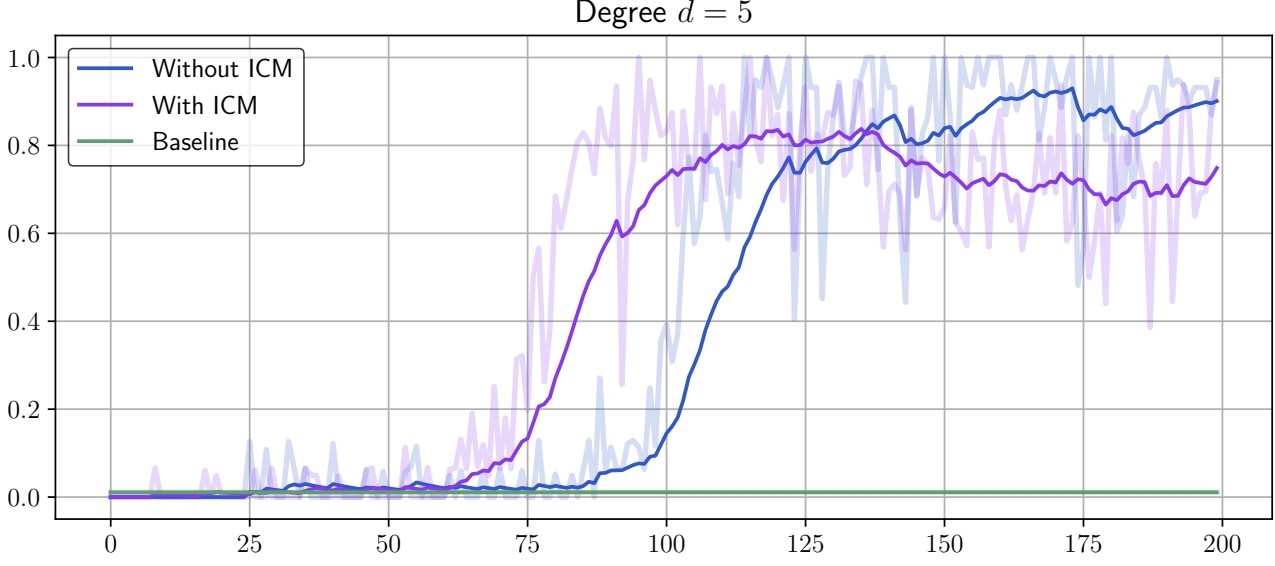

*Figure 14.* Average episodic return of HRL agent with and without ICM module.

### E. New Non-Hirsch Ideals From Old

In this section we describe another direction we have explored for constructing new non-Hirsch ideals by using the list of existing ones. This complements other approaches described in the main text that mainly focus on constructing non-Hirsch ideals of a fixed degree $d$ in a polynomial ring $k[X_n]$ with a fixed number of variables $X_n := \{x_1, \ldots, x_n\}$ without prior knowledge of other non-Hirsch ideals.

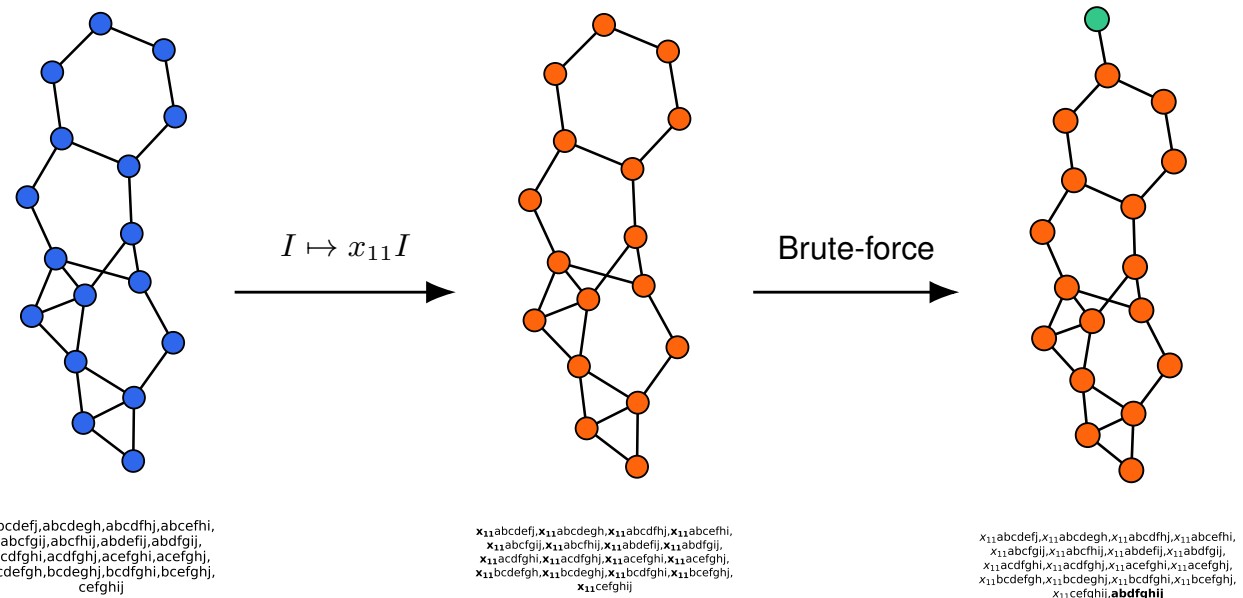

abcdefj,abcdegh,abcdfhj,abcefhi,
abcfgij,abcfhij,abdefij,abdfgij,
acdfghi,acdfghj,acefghi,acefghj,
bcdefgh,bcdeghj,bcdfghi,bcefghj,
cefghij

$x_{11}$abcdefj,$x_{11}$abcdegh,$x_{11}$abcdfhj,$x_{11}$abcefhi,
$x_{11}$abcfgij,$x_{11}$abcfhij,$x_{11}$abdefij,$x_{11}$abdfgij,
$x_{11}$acdfghi,$x_{11}$acdfghj,$x_{11}$acefghi,$x_{11}$acefghj,
$x_{11}$bcdefgh,$x_{11}$bcdeghj,$x_{11}$bcdfghi,$x_{11}$bcefghj,
$x_{11}$cefghij

$x_{11}$abcdefj,$x_{11}$abcdegh,$x_{11}$abcdfhj,$x_{11}$abcefhi,
$x_{11}$abcfgij,$x_{11}$abcfhij,$x_{11}$abdefij,$x_{11}$abdfgij,
$x_{11}$acdfghi,$x_{11}$acdfghj,$x_{11}$acefghi,$x_{11}$acefghj,
$x_{11}$bcdefgh,$x_{11}$bcdeghj,$x_{11}$bcdfghi,$x_{11}$bcefghj,
$x_{11}$cefghij,**abdfghij**

*Figure 15.* Process of constructing new Non-Hirsch ideals from old.

Let $I$ be a non-Hirsch ideal of degree $d$ in $k[X_n]$ generated by monomials $\{f_i\}_{i=1}^{g}$. We may embed this ideal into a polynomial ring $k[X_n][x_{n+1}] = k[X_{n+1}]$ by multiplying each generator of $I$ by $x_{n+1}$:

$$f_i \longmapsto x_{n+1}f_i\,. \tag{28}$$

This produces an ideal $\tilde{I} = x_{n+1}I$ of degree $d + 1$ in $k[X_{n+1}]$ which preserves both the diameter and the number of irreducible edges. Since $I$ is non-Hirsch and therefore linearly presented, we must have $\tilde{I}$ also linear. What remains to do is to attach another generator to the ideal $\tilde{I}$ in order to increase the diameter to $d + 2$ whilst ensuring that no new irreducible edges are introduced by this operation. While the number of possible new generators that may be added to the ideal $\tilde{I}$ grows combinatorially, we find that a simple brute-force search is sufficient.

