# OpenReview forum: "Hierarchical Reinforcement Learning for Sparse-Reward Search in Commutative Algebra"
_ICML.cc/2026/Conference — ICML 2026 regular_

### Official Review · Reviewer_8V8f · 2026-03-10

**Soundness:** 4
**Presentation:** 3
**Significance:** 3
**Originality:** 3
**Overall Recommendation:** 5
**Confidence:** 2

**Summary:**

They proposed a HRL framework for lgebraic Hirsch Conjecture which i am not familiar with. What i get is authors formulate the mathematical problem into a seach problem. They slove this problem by traditional search and reinforcement learning.

**Compliance With Llm Reviewing Policy:**

Affirmed.

**Final Justification:**

This study aims to apply hierarchical reinforcement learning method for algebra problem. As i see, this is a good paper in origniality and presentation. Based on the paper and author's response, i appreicate this is a good topic for discssion. I keep my socre. Thanks.

**Key Questions For Authors:**

How to evaluate the correcntness ?

What's the impact from sloving this math problem?

The comparsion with human sloving can be provide?

**Limitations:**

Disscusion on the generalization of the propose method.
More experiment results on the correctness of problem sloving.
The method can be compared between PPO, AC, etc

**Strengths And Weaknesses:**

The problem was sloved soundly nice.

the presentation is also good.

They use RL to slove the mathematical hard problem, althought i am not sure where the problem can be used for.

It has a good originality.

---

> ### Author Rebuttal · Authors · 2026-03-31
>
> We sincerely thank the reviewer for carefully reading our manuscript and for their thoughtful questions. Below, we provide answers to the questions raised in the review.
> ___
> ## Q1: How to evaluate correctness?
> The environment interactions correspond to well-defined operations on ideals. Therefore, correctness is **fully verifiable** at each step. There are two steps involved in verifying the correctness:
>
> 1.  **Large Diameter Condition**: Given a state corresponding to an ideal $I$, we need to verify that it satisfies the large diameter condition: $\mathrm{diam}(I)>d$. In order to compute the diameter, we apply Floyd-Warshall algorithm, or Dijkstra, depending on the size of the graph. This allows us to compute the shortest distance matrix, whose maximum yields the diameter.
> 2. **Linearity Condition**: Similarly, given that each state corresponds to an ideal $I$, we need to verify that there are no irreducible edges (as defined in *Def. A.5*). For this, we employ the algorithm described in Appendix A.
>
> Both conditions can be checked exactly (not heuristically). Therefore, any solution constructed using RL is **provably correct**.
>
> ## Q2: What is the impact from solving this math problem?
>
> ### (Math Impact):
> In commutative algebra, understanding linear syzygies is a very central problem in many areas. These linear syzygies encode subtle geometric information, and remain poorly understood even in classical settings. Finding non-Hirsch ideals has been a significant challenge, with only a single example at $d=7$ known prior to this work. Our approach successfully generates such ideals at scale, demonstrating that appropriately designed hierarchical reinforcement learning can meaningfully contribute to pure mathematics research.
> To our knowledge, this represents the first application of RL to commutative algebra, opening new avenues for AI-assisted mathematical discovery.
>
> ### (RL Impact)
> In RL, the key impact lies in demonstrating the capabilities of suitable hierarchical RL algorithms to extremely sparse reward problems, such as the ones arising from problems in pure mathematics. The success of our approach shows that domain-informed temporal abstractions with constraints can be essential when reward signals are vanishingly sparse. Our approach has the potential to be particularly well-suited to problems with two key properties:
>
> 1. **Unknown goals**, where the goal state is not a priori known and must itself be constructed.
> 2. **Many local maxima**, where the task can be decomposed into identifying a representative state and then reaching the target.
>
> The behavioral analysis that we describe in section 6 shows that the identification of bottleneck states can be successfully applied to such "needle-in-a-haystack"-like problems.
>
> ## Q3: How does this compare to a human solver?
> There are currently no established human benchmarks or hand-crafted algorithms for constructing non-Hirsch ideals. During our work, we developed both the classical (i.e. non-ML) algorithmic approaches, namely greedy search (GS), and standard RL methods, such as PPO and SAC using GNNs.
>
> Within these baselines, GS provides the strongest prior baseline, which outperforms classical RL methods. Our proposed HRL approach, utilizing chained constrained options, consistently outperforms GS across all degrees we tested it on.
> ___
> We will clarify these points and expand discussion in the revision.

---

> > ### Author Rebuttal · Reviewer_8V8f · 2026-04-01
> >
> > I agree with author's response. Hence, i will keep my score, as this question is a good topic for discussion.

---

### Official Review · Reviewer_sCic · 2026-03-11

**Soundness:** 3
**Presentation:** 3
**Significance:** 2
**Originality:** 3
**Overall Recommendation:** 5
**Confidence:** 2

**Summary:**

This paper recasts a commutative algebra problem into a sparse-reward RL problem on graphs. Authors propose a constrained options-based HRL framework with an equivariant graph neural network policy and demonstrate its effectiveness over classical RL algorithms as well as greedy search.

**Compliance With Llm Reviewing Policy:**

Affirmed.

**Final Justification:**

Updated due to rebuttal

**Key Questions For Authors:**

My questions are based on concerns in the "weaknesses" part.

1. Is it able to compare the proposed method with baseline graph-based HRL methods?

2. Does the method have the potential to be applied to other Math or more general tasks?

**Limitations:**

Yes.

**Strengths And Weaknesses:**

Strengths:

The theoretical foundation and the algorithm are very clearly explained. The writing is clear as well.

Weaknesses:

1. This paper may benefit from comparing with other graph-based / HRL methods. (such as [1] and [2])

[1] Klissarov, Martin, and Doina Precup. "Reward propagation using graph convolutional networks." Advances in Neural Information Processing Systems 33 (2020): 12895-12908.

[2] Lee, Seungjae, et al. "Dhrl: A graph-based approach for long-horizon and sparse hierarchical reinforcement learning." Advances in neural information processing systems 35 (2022): 13668-13678.

2. The scope of this work is a little bit limited. Authors may want to explore other similar tasks with the proposed approach to demonstrate its generalizability.

---

> ### Author Rebuttal · Authors · 2026-03-31
>
> We thank the reviewer for providing constructive feedback and suggesting other graph-based HRL methods. We address the questions mentioned in the review below.
> ___
>
> ## Q1: Is it able to compare the proposed method with baseline graph-based HRL methods?
> We explored several related graph-based and goal-conditioned approaches (including HER, TER (Topological Experience Replay), PHIL (Path Heuristic with Imitation Learning) and Option-Critic), but did not observe any measurable improvements over standard PPO/SAC baselines. Therefore, these results were omitted due to space constraints. We will include a brief summary of these results in the revised version.
>
> Regarding cited works:
>
> * **Reward propagation using GCNs** [1]: This approach is most similar to TER as both leverage the graph structure to enable non-local credit assignment in sparse-reward settings. However, without clear constrained options, small local modifications to an ideal can cause discontinuous changes in the global properties (e.g., diameter). Therefore, neighboring states may provide weak or misleading signals for reward propagation.
> * **DHRL** [2]: This approach assumes (i) clear goal space and (ii) well-defined distance metric between states. Both assumptions are difficult to satisfy in our setting due to the following reasons:
>     1. **No explicit goal space**. The valid solutions (non-Hirsch ideals) are not known a priori and therefore cannot be used as goals. We have attempted using indirect goals, such as the tuple $(\mathrm{Diam}(I)-d,~ \mathrm{\\#irr}(I))$ of two required properties for an ideal to be non-Hirsch. These conditions measure how "far" the ideal is from being non-Hirsch. However, due to strong non-linearity, these signals did not provide a useful notion of intermediate goals.
>     2. **Lack of metrizability**. In our setting, environment transitions are highly non-smooth: small changes in the state can lead to large, global changes in the algebraic properties of the ideal. As a result, constructing such distance metric is equally difficult as solving the problem.
>
> More broadly, DHRL excels in environments with **locally smooth** dynamics (e.g., navigation). However, our environment is highly **non-local and combinatorial**, which limits the effectiveness of such approaches.
>
> In contrast, our approach relies on exploiting emergent bottleneck structure identified using $f_\pi(n)$ (Eq. 7), which led us to define constrained options that construct specific line-graph ideals ("spines") contained in non-Hirsch ideals.
>
>
> ## Q2: Does the method have the potential to be applied to other Math or more general tasks?
>
> Yes. Mathematical reasoning, both for constructing proofs and finding examples/counterexamples, is inherently hierarchical. A similar two-stage structure also appears in many mathematical search problems, such as problems in number theory. For example, sieve and distributions methods already give bounded gaps between primes and prime-rich admissible tuples, but the parity problem says classical sieve methods cannot reliably distinguish "an odd number of prime factors" from "an even number of prime factors". This creates a fundamental obstruction: existing methods can reach an **"almost-solution" regime** (e.g, almost-primes or bounded gaps), but fail at enforcing the final exact constraint required for the results, such as twin primes. This illustrates a two-step hierarchical pattern, where one method handles the **coarse structure** and a fundamentally different method addresses the **exact/global constraint**. Similar two-stage decompositions arise in problems such as the Goldbach conjecture, Diophantine approximation vs Diophantine equations, etc.
>
> Our framework is designed precisely for such settings, where the hierarchical decomposition (via bottleneck states constructed via constrained interactions) separates the construction of promising intermediate objects from the enforcement of global properties. Although our work focuses on a specific algebraic setting, we expect this approach to extend to other combinatorial search problems exhibiting similar two-stage decomposition. For more detailed breakdown of the key properties required by our framework, please see our response to Q2 (RL Impact) to the reviewer 8V8f.
>
> We will include a more detailed discussion of generalizability in the revised version.

---

> > ### Author Rebuttal · Reviewer_sCic · 2026-04-01
> >
> > Thank you for your response. I believe most of my concerns have been addressed, and I will be updating my score accordingly.

---

### Official Review · Reviewer_Ng6e · 2026-03-12

**Soundness:** 3
**Presentation:** 3
**Significance:** 3
**Originality:** 3
**Overall Recommendation:** 5
**Confidence:** 1

**Summary:**

This paper introduces a Hierarchical Reinforcement Learning (HRL) framework designed to solve extreme reward sparsity in commutative algebra, specifically for finding counterexamples to Kalai’s algebraic Hirsch conjecture. The authors utilize a constrained options-based approach and an equivariant graph neural network to navigate complex combinatorial search spaces and identify rare "non-Hirsch" monomial ideals. Their methodology consistently outperforms standard RL algorithms and greedy searches, successfully scaling to larger degrees where traditional methods fail.

**Compliance With Llm Reviewing Policy:**

Affirmed.

**Key Questions For Authors:**

Given the extreme reward sparsity, how did you determine that the "spine" structure was the most effective temporal abstraction to use for the high-level policy?

Do you believe this hierarchical approach can be adapted to find counterexamples for other conjectures in commutative algebra, or is it specifically tailored to the Hirsch conjecture?

**Limitations:**

Yes

**Strengths And Weaknesses:**

StrengthsSoundness:

The empirical evaluation is rigorous, comparing the HRL approach against strong baselines like PPO, SAC, and Breadth-First Search across multiple degrees ($d$). The use of "spines" as a structural prior is well-supported by an analysis of successful trajectories, and the implementation of a Syzygy-aware GNN (SConv) appropriately addresses the specific algebraic properties of monomial ideals.

Originality: The paper presents a highly creative application of RL to a specialized field of pure mathematics (commutative algebra). The "Chained Constrained Options" framework is a novel way to encode mathematical constraints directly into the MDP, effectively bridging the gap between symbolic AI and deep reinforcement learning.

Significance: This work represents a breakthrough in using AI to attack long-standing conjectures (Kalai’s algebraic Hirsch conjecture). By successfully finding counterexamples where standard RL fails, it provides a template for how HRL can be used to navigate search spaces with extreme reward sparsity, which has implications for both automated theorem proving and combinatorial optimization.

Presentation: The paper is well-structured, clearly defining complex mathematical concepts (like Betti numbers and linear presentations) for a machine learning audience. The visualization of the "spine" construction and the ablation studies help clarify how each component contributes to the agent's success.

Weaknesses:

Presentation: Some of the technical details regarding the SConv (Syzygy-aware) layer and the exact hyperparameter tuning for the hierarchical rewards are dense and may be difficult for readers without a background in both GNNs and abstract algebra to fully reproduce without the provided code.

---

> ### Author Rebuttal · Authors · 2026-03-31
>
> We thank the reviewer for a thorough review of our submission. Below we address the questions raised in the review.
> ___
> ## Q1: Given the extreme reward sparsity, how did you determine that the "spine" structure was the most effective temporal abstraction to use for the high-level policy?
>
> After evaluating the interaction of the agent trained using regular PPO and SAC (Sec. 5) for the lowest possible non-trivial degree $d=4$, we identified the least topologically complex graph encountered in each trajectory. The idea being, if there was a common type of a graph encountered along the trajectory, it would be possible to leverage it to improve exploration efficiency. For this reason, we picked the number of cycles $b_1(G)$ as a measure of the "complexity" of $G$ (See Eq. 7). We then noticed that the graph with no cycles appeared to be the most common type of graph encountered (See Fig. 4).
>
> While trees also fall within this class of graphs, we restricted our attention to focus solely on constructing line graphs for the following reasons:
>
> * **Necessity of spines**: Any non-Hirsch ideal contains a diameter-maximizing line-graph as its subgraph.
> * **Clear lower bound on the horizon**: We must take at least $d+1$ steps in order to construct a spine. For trees, this may depend on branches, which can potentially increase the required episode length.
> * **Reduction of the number of actions**: The number of allowed actions is significantly smaller and limited to nodes that can be attached to the end points.
>
> ## Q2: Do you believe this hierarchical approach can be adapted to find counterexamples for other conjectures in commutative algebra, or is it specifically tailored to the Hirsch conjecture?
>
> Yes. While our approach focuses on the Algebraic Hirsch Conjecture, the general two-step constrained hierarchical framework is not specific to this problem and we expect it to be generalizable to a broader class of problems in commutative algebra and beyond.
>
> ### Application to math problems:
> As discussed in our responses to Q2s to the reviewers sCic and 8V8f, our approach has the potential to be particularly well-suited to problems with two key properties (i) unknown goals, where goal state is not known beforehand and (ii) many local maxima, where task can be decomposed into identifying a suitable promising region of the search space and then reaching the goal state.
>
> This pattern appears in many different math problems, for example the unknotting problem in topology, where the work of [1] on random unknotting demonstrates that solution difficulty is highly dependent on the initial presentation, with "hard" presentations requiring fundamentally different search strategies: precisely the regime our spine-based approach addresses.
>
> ### Relevance within commutative algebra:
> The study of linear syzygies is central to many problems in algebraic geometry and commutative algebra, such as Green's conjecture, which relates linear syzygies of a curve to a global geometric invariant (the Clifford index).
>
> Generally, many problems in commutative algebra require constructing algebraic objects satisfying global constraints, which are difficult to enforce through local operations. This naturally leads to a decomposition into a **coarse construction stage**, where one builds intermediate objects with almost-correct structure (e.g., spines), followed by a second, fundamentally different method for enforcing the **global constraint**.
>
> ## References
> [1] J. Cantarella, et al., Random knotting in very long off-lattice self-avoiding polygons. Journal of Physics A 2026

---

> > ### Author Rebuttal · Reviewer_Ng6e · 2026-04-02
> >
> > Thank you for your response. I believe most of my concerns have been addressed

---

### Official Review · Reviewer_orGB · 2026-03-13

**Soundness:** 3
**Presentation:** 3
**Significance:** 3
**Originality:** 3
**Overall Recommendation:** 5
**Confidence:** 2

**Summary:**

This paper tackles the challenge of finding counterexamples for the Algebraic Hirsch Conjecture, a task characterized by extreme reward sparsity and a combinatorially explosive search space. The authors propose a Hierarchical Reinforcement Learning framework using Chained Constrained Options.

**Compliance With Llm Reviewing Policy:**

Affirmed.

**Final Justification:**

Good work. The authors' response have addressed my remaining concerns. The only issue is that I'm not very familiar with this domain thus I don't have a strong confidence of my evaluation.

**Key Questions For Authors:**

See weakness above

**Limitations:**

Yes

**Strengths And Weaknesses:**

**Strengths**:

1. The authors studied an important mathematics problem using hierarchical reinforcement learning. They successfully designed an HRL algorithm to search for the counterexamples for the Algebraic Hirsch Conjecture.

2. The writing of this paper is very clear and easy to follow.

**Weakness**:

1. This paper lacks some related works, including comparison with other learning based works for Algebraic Hirsch Conjecture and comparison with other HRL methods for Science.

2. Is there any other baselines for searching the counterexamples by using searching algorithms like Monte Carlo Tree Search or SAT solver?

3. Is the method developed in this work time consuming?

---

> ### Author Rebuttal · Authors · 2026-03-31
>
> We thank the reviewer for detailed review and constructive feedback. We address some of the concerns raised in the review below.
> ___
> ## Q1: Related works
> To our knowledge, there are no previously established learning or HRL methods for constructing non-Hirsch ideals. Our work is therefore first to formulate this problem as a two-step hierarchical RL task.
>
> We explored several HRL and goal-based RL algorithms (e.g., Option-Critic, HER, TER), but did not observe any improvements over standard RL baselines. Please see our response to Q1 to the reviewer sCic for a more detailed discussion on other graph-based HRL techniques. We will include a concise comparison and discussion of these approaches in the revised version.
>
> ## Q2: Is there any other baselines for searching the counterexamples by using searching algorithms like MCTS or SAT solver?
> While developing our baselines, we have explored both MCTS and SAT solvers. Below we discuss more details regarding our findings:
>
> * **MCTS**: We implemented a standard MCTS baseline in our environment. However, due to the extremely sparse terminal reward signal and large branching factor, MCTS struggled to make meaningful progress. In particular, rollouts were dominated by per-step penalties, and the visit distribution at the root remained low-confidence. In our preliminary experiments ($d=4$ with 50 simulations and rollout depth of 100), MCTS did not reliably discover terminal states or solutions.
> * **SAT solvers**: We also explored SAT-based approaches. While SAT solvers are well-suited for the problems which may be expressed in terms of local properties, our setting involves two global structural properties:
>     * **Diameter**, which depends on all paths in the graph and is difficult to encode efficiently in CNF form.
>     * **Linearity**, which also similarly depends on existence of certain restricted paths between the endpoints of edges in $\cup_{l>1}E_l$.
>
> While there might be some ways to circumvent this issue, it is unclear and highly non-trivial which encoding of global constraints would achieve this goal. In particular, it would require:
> * A tractable number of clauses.
> * An efficient algorithm for finding the necessary clauses.
>
> That being said, SAT solver did have some success for a similar *realizability* problem, where a graph topology is given and the question is to find its realization as an ideal. Given each candidate graph, the problem is to assign degree $d$ monomials in $n$ variables to each node, such that two nodes share an edge if and only if their monomials differ in exactly 2 variables (Hamming distance 2). We used Glucose3 (PySAT) solver. However, this approach does not address the primary search problem of discovering non-Hirsch ideals, and in our experiments, it did not lead to any successful constructions.
> ## Q3: Is the method developed in this work time consuming?
> All RL agents in our paper are trained on a single GPU. While the training time naturally increases with degree because of both increased reward sparsity and larger number of allowed actions, we do not find the method to be prohibitively time consuming.
>
> At *inference time*, generating a new ideal remains fast: for the largest degree considered in our paper, the trained agent produces a new ideal in approximately **10 seconds** on a single RTX 3090 GPU.
>
> As for the training dynamics, the point at which the mean episodic return sharply increases shifts roughly linearly from 10 to 350 steps as the degree increases from $d=4$ to $d=7$. Furthermore, each environment interaction costs roughly $1/48$ GPU seconds across all degrees. This is primarily due to the following reasons:
> * **Efficient Linearity Checks**: Our custom CUDA kernels offer $\bf 50\times$-$\bf 100\times$ speedup for linearity checks (See Fig. 9 in Appendix A), depending on the degree and codimension. Furthermore, in all cases we have observed, it is sufficient to check reducibility of edges $E_2$, rather than all high-level edges $\cup_{l>1}E_l$.
> * **Linear path growth**: The mean path length after constructing a spine grows roughly linearly from 4 to 14 as the degree increases from $d=4$ to $d=7$.
> | **Quantity**         | $d=4$ | $d=5$ | $d=6$ | $d=7$ |
> | ---------------- | --- | --- | --- | --- |
> | Path length$^{*}$ | $4.3\pm 0.5$   | $8.3\pm 1.5$  | $10.7\pm 2.5$  | $14.4\pm 3.24$  |
> | Max path length | $9$ | $14$ | $21$ | $24$ |
> *$^{\*}$ $\pm$ denotes an empirical interval containing 95% of sampled trajectories. Max path length indicates maximum observed path length over all trajectories ever sampled.*
> * **Linear episode scaling**: The length of a spine, which is essentially equal to the episode horizon, scales roughly linearly with degree. This is because a spine of diameter $d+1$ needs only $d+2$ nodes.
>
> Overall, both the empirical scaling trends and the GPU-accelerated verifier timing benchmarks indicate that our proposed method remains computationally practical, even at the highest degrees studied.

---

> > ### Author Rebuttal · Reviewer_orGB · 2026-04-01
> >
> > Thanks for the response. I think my concerns are mostly addressed. I will raise my score.

---

### Decision · Program_Chairs · 2026-04-30

**Decision:**

Accept (regular)

**Comment:**

All reviewers appreciated the contribution and agreed in favor of acceptance.